# Identification of the bacteriophage nucleus protein interaction network

Eray Enustun[1], Amar Deep [2], Yajie Gu[2], Katrina T. Nguyen[1], Vorrapon Chaikeeratisak [1,3], Emily Armbruster [1], Majid Ghassemian[4], Elizabeth Villa [1,5], Joe Pogliano [1] ✉ & Kevin D. Corbett [1,2] ✉

In the arms race between bacteria and bacteriophages (phages), some large-genome jumbo phages have evolved a protein shell that encloses their replicating genome to protect it against host immune factors. By segregating the genome from the host cytoplasm, however, the 'phage nucleus' introduces the need to specifically translocate messenger RNA and proteins through the nuclear shell and to dock capsids on the shell for genome packaging. Here, we use proximity labeling and localization mapping to systematically identify proteins associated with the major nuclear shell protein chimallin (ChmA) and other distinctive structures assembled by these phages. We identify six uncharacterized nuclear-shell-associated proteins, one of which directly interacts with self-assembled ChmA. The structure and protein–protein interaction network of this protein, which we term ChmB, suggest that it forms pores in the ChmA lattice that serve as docking sites for capsid genome packaging and may also participate in messenger RNA and/or protein translocation.

Since the discovery of phages more than a century ago, research on these remarkable entities has yielded fundamental insights into a broad range of pathways across biology[1]. Historically, most phage studies have focused on small-genome phages (~30–140 kb), leaving larger 'jumbo phages' with genomes over ~200 kb much less well understood despite their abundance in nature[2–4]. We previously showed that one family of jumbo phages forms distinctive structures in infected cells, including a nucleus-like compartment bounded by a proteinaceous shell and a spindle-like structure that centers and rotates this compartment within the host cell[5–7]. The phage nucleus encloses the replicating phage genome and excludes most host proteins, including CRISPR effectors and restriction enzymes, rendering this family of phages broadly resistant to DNA-targeting bacterial immune systems[8,9].

The jumbo phage nuclear shell provides an important selective advantage by protecting the replicating phage genome from host-encoded defense nucleases, but that protection comes at the cost of substantial added complexity in the phage life cycle. The phage nuclear shell is composed primarily of one protein, termed chimallin (ChmA) or phage nuclear enclosure (PhuN), which forms a single-layer-thick flexible lattice that separates the phage genome from the bacterial cytoplasm[10,11]. Pores in the ChmA lattice are less than ~2 nm in width, large enough to pass metabolites but too small for passage of most proteins or messenger RNAs (mRNAs)[10]. As in eukaryotes, mRNAs are transcribed within the phage nucleus but translated in the cytoplasm, meaning that phage mRNAs must be translocated out of the nucleus[6]. At the same time, phage-encoded proteins necessary for genome replication and mRNA transcription must be specifically translocated into the nucleus[6,12]. Finally, during virion production, newly assembled capsids are trafficked along filaments of the tubulin-like protein PhuZ to the nuclear shell, where they dock for genome packaging[6,13]. Following genome packaging, capsids are assembled with virion tails at a pair of structures termed the 'phage bouquets' before cell lysis and virion release[14,15].

[1]Department of Molecular Biology, University of California San Diego, La Jolla, CA, USA. [2]Department of Cellular and Molecular Medicine, University of California San Diego, La Jolla, CA, USA. [3]Department of Biochemistry, Faculty of Science, Chulalongkorn University, Bangkok, Thailand. [4]Biomolecular and Proteomics Mass Spectrometry Facility, University of California San Diego, La Jolla, CA, USA. [5]Howard Hughes Medical Institute, La Jolla, CA, USA. ✉e-mail: jpogliano@ucsd.edu; kcorbett@ucsd.edu

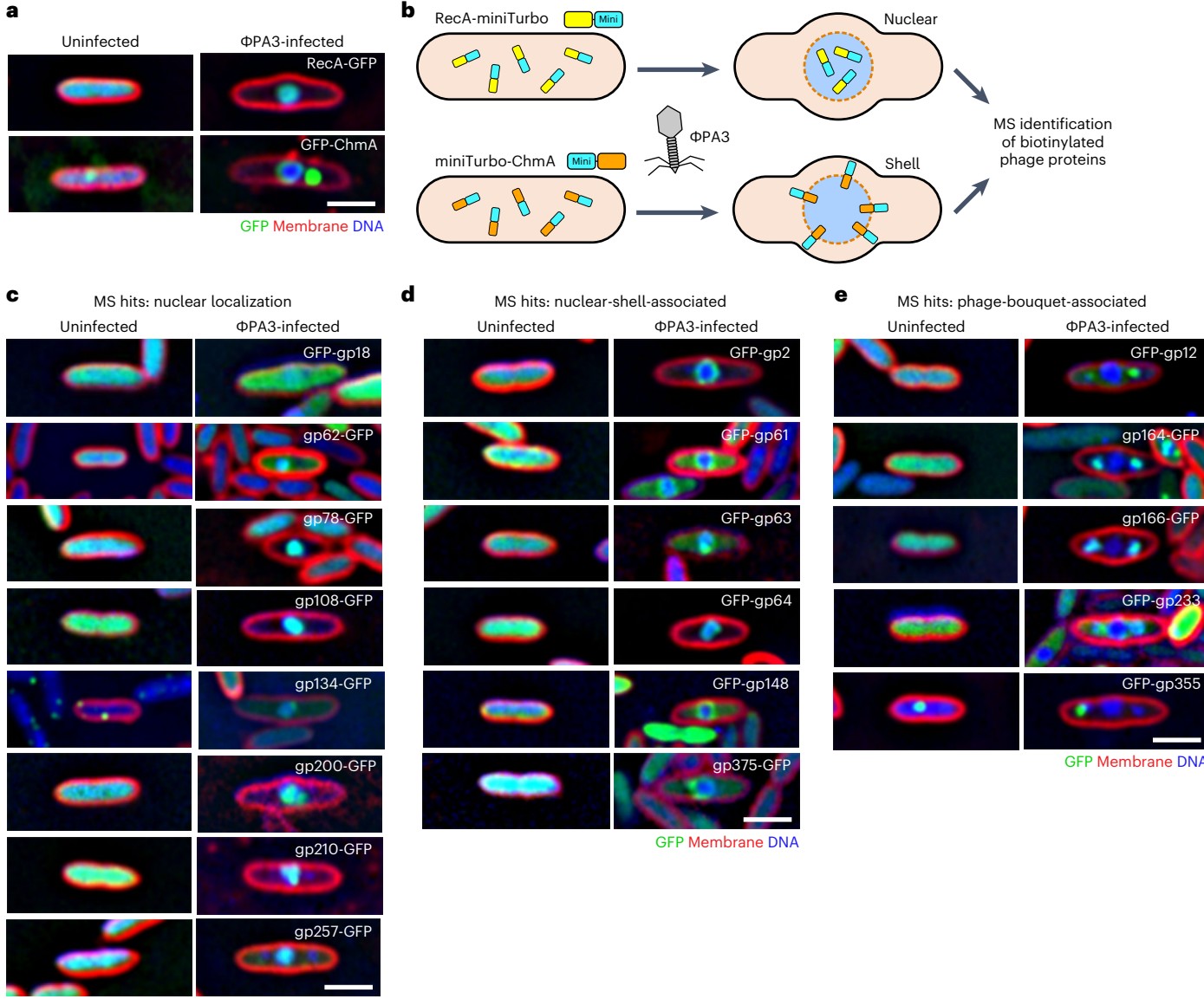

**Fig. 1 | Identification of jumbo phage nuclear-shell-associated proteins. a**, Subcellular localization of GFP-tagged ΦPA3 RecA (gp175) and ChmA (gp53) in uninfected (left) and ΦPA3-infected (right) *P. aeruginosa* cells. GFP is shown in green, FM4-64 (to visualize membranes) in red and DAPI (to visualize nucleic acids) in blue. Scale bar, 2 μm. **b**, Experimental schematic for identification of jumbo phage nuclear or nuclear-shell-associated genes by proximity labeling with miniTurboID-fused RecA (C-terminal miniTurboID) or ChmA (N-terminal miniTurboID) in ΦPA3-infected *P. aeruginosa* cells. See Extended Data Fig. 1a for fusion construct design; Extended Data Fig. 1b for localization of miniTurboID-

fused proteins; Tables 1 and 2 for the top 25 identified proteins; Supplementary Tables 1 and 2 for full protein lists; and Extended Data Fig. 1c,d for diagrams showing overlap between independent mass spectrometry datasets. **c–e**, Subcellular localization of selected proteins identified by proximity labeling, showing nuclear-localized proteins (**c**), nuclear-shell-associated proteins (**d**) and phage bouquet-associated proteins (**e**). See Extended Data Fig. 2 for further data and Supplementary Table 3 for a collated list of localizations. Scale bar, 2 μm. MS, mass spectrometry.

The diverse functions of the jumbo phage nuclear shell—including mRNA and protein translocation through the shell and capsid docking on the shell—imply that this structure incorporates multiple components in addition to ChmA that mediate these functions. Here, we use proximity labeling (miniTurboID[16]) in *Pseudomonas aeruginosa* cells infected by the jumbo phage ΦPA3 (ref. 17) to identify proteins that localize both within the phage nucleus and specifically to the nuclear shell. We identify six new nuclear-shell-associated proteins, one of which interacts directly with both ChmA and the putative portal protein[18]. These interactions suggest that this protein, which we term ChmB, forms pores in the ChmA lattice and mediates capsid docking and genome packaging. The overall protein–protein interaction

network of ChmB further suggests additional roles in mRNA and/or protein translocation across the phage nuclear shell. More broadly, our data define the protein interaction network of the jumbo phage nuclear shell and reveal the subcellular localization of dozens of previously uncharacterized jumbo phage proteins.

## Results

### Identification of proteins associated with the jumbo phage nucleus

The genomes of nucleus-forming jumbo phages are poorly characterized: for instance, 290 of the 378 genes encoded by the *Pseudomonas* jumbo phage ΦPA3 have no annotated function in the NCBI protein database.

**Table 1 | Top 25 identified proteins from ΦPA3 ChmA (gp53) miniTurboID**

| Protein | NCBI accession number | Times detected in three trials | log$_2$ (fold change)[a] | Normalized peak area[b] | Annotation | Found in RecA miniTurbo ID? |
|---|---|---|---|---|---|---|
| gp53 (ChmA) | YP_009217136.1 | 3 | 4.00 | $3.25×10^{-2}$ | Major nuclear shell protein | |
| gp52 | YP_009217135.1 | 3 | 3.27 | $2.09×10^{-2}$ | Hypothetical protein | |
| gp355 | YP_009217434.1 | 3 | 3.79 | $6.49×10^{-3}$ | Hypothetical protein | |
| gp164 | YP_009217199.1 | 3 | 3.93 | $1.04×10^{-3}$ | Tail protein | |
| gp119 | YP_009217199.1 | 3 | ∞ | $8.81×10^{-4}$ | Hypothetical protein | Yes |
| gp166 | YP_009217245.1 | 2 | ∞ | $8.75×10^{-4}$ | Tail protein | Yes |
| gp358 | YP_009217437.1 | 3 | 8.51 | $6.90×10^{-4}$ | Hypothetical protein | Yes |
| gp2 | YP_009217084.1 | 2 | ∞ | $5.87×10^{-4}$ | Hypothetical protein | |
| gp134 | YP_009217213.1 | 3 | ∞ | $5.00×10^{-4}$ | Putative helicase | Yes |
| gp131 | YP_009217211.1 | 2 | ∞ | $3.45×10^{-4}$ | Putative helicase | Yes |
| gp63 | YP_009217146.1 | 2 | ∞ | $2.85×10^{-4}$ | Hypothetical protein | Yes |
| gp62 | YP_009217145.1 | 2 | ∞ | $2.51×10^{-4}$ | nvRNAP subunit | Yes |
| gp222 | YP_009217301.1 | 2 | ∞ | $2.13×10^{-4}$ | Hypothetical protein | Yes |
| gp106 | YP_009217186.1 | 2 | 6.10 | $1.57×10^{-4}$ | Virion structural protein | |
| gp18 | YP_009217100.1 | 1 | ∞ | $1.57×10^{-4}$ | Hypothetical protein | Yes |
| gp202 | YP_009217281.1 | 1 | ∞ | $1.55×10^{-4}$ | Hypothetical protein | |
| gp148 | YP_009217227.1 | 1 | ∞ | $1.20×10^{-4}$ | Portal protein | |
| gp14 | YP_009217096.1 | 1 | 4.95 | $1.08×10^{-4}$ | Hypothetical protein | |
| gp12 | YP_009217094.1 | 1 | ∞ | $9.40×10^{-5}$ | Tail protein | Yes |
| gp378 | YP_009217457.1 | 2 | ∞ | $8.48×10^{-5}$ | NrdA | |
| gp247 | YP_009217326.1 | 1 | ∞ | $4.45×10^{-5}$ | Hypothetical protein | Yes |
| gp257 | YP_009217336.1 | 2 | 3.21 | $3.90×10^{-5}$ | DNA ligase | Yes |
| gp219 | YP_009217298.1 | 1 | ∞ | $3.87×10^{-5}$ | Hypothetical protein | |
| gp335 | YP_009217414.1 | 1 | ∞ | $3.83×10^{-5}$ | Hypothetical protein | Yes |
| gp85 | YP_009217165.1 | 1 | ∞ | $3.67×10^{-5}$ | Hypothetical protein | |
| gp370 | YP_009217449.1 | 1 | ∞ | $2.79×10^{-5}$ | Hypothetical protein | |

[a]Fold change calculated as the fold change in average normalized peak area when comparing three trials with three negative-control trials. [b]Normalized peak area calculated as the fraction of the total peak area for ΦPA3 proteins (per dataset) assigned to a given protein.

To overcome this deficit, we used a proximity labeling approach to identify proteins associated with the phage nuclear shell that could endow this structure with additional functionality such as mRNA or protein translocation and capsid docking. We fused the promiscuous biotin ligase miniTurboID[16] to the ChmA protein from the jumbo phage ΦPA3 (gp53) and to the phage's nuclear-localized RecA protein (gp175)[7] (Fig. 1a,b and Extended Data Fig. 1a,b). We first verified that fusing either gp53 or gp175 to green fluorescent protein (GFP) and miniTurboID did not alter the localization of these proteins in ΦPA3-infected cells (Extended Data Fig. 1a,b). We then expressed miniTurboID-gp53 or gp175-miniTurboID (lacking GFP to minimize off-target effects) in *P. aeruginosa* cells infected with ΦPA3, collected samples at 45 min postinfection, which is the earliest time point at which the mature nuclear shell is observed with docked capsids[7], and performed streptavidin pulldown and mass spectrometry analysis to identify biotinylated proteins. By focusing on the phage proteins that were biotinylated and normalizing the results to a control miniTurboID-GFP fusion (which remains diffuse in the host cell cytoplasm throughout infection; Extended Data Fig. 1b), we identified candidate proteins that preferentially localize in close proximity to ChmA and/or RecA (Tables 1 and 2 and Supplementary Tables 1 and 2).

To validate our interaction data, we generated GFP fusions of the top 25 ChmA-interacting and RecA-interacting proteins from our miniTurboID datasets (42 proteins plus ChmA and RecA; Supplementary Table 3), expressed each in *P. aeruginosa* and determined their localization in both uninfected and ΦPA3-infected cells. Proteins that interact with RecA are expected to localize inside the phage nucleus, whereas ChmA-interacting proteins are expected to localize on or near the nuclear shell. Given the architecture of the ChmA lattice, we expected the amino-terminal miniTurboID tag to be localized on the outer surface of the nuclear shell[10]. Among the 42 proteins tested, we identified eight that localized within the nucleus-like compartment, six that localized to the nuclear shell itself and five that localized to the phage bouquets (Fig. 1c–e). Most proteins that localized within the phage nucleus were annotated in the NCBI database as putative nucleic-acid-interacting proteins, including a subunit of the phage-encoded nonvirion RNA polymerase (nvRNAP; gp62)[19], two predicted helicases (gp131 and gp134), a predicted DNA ligase (gp257) and two predicted endonucleases (gp78 and gp210)[18] (Fig. 1c and Supplementary Table 3). Two nuclear-localized proteins (gp108 and gp200) had no annotated or predicted function. Of the five proteins that localized to phage bouquets, three (gp12, gp164 and gp166) were predicted phage tail proteins, one (gp233) was a predicted helicase and one (gp355) had no annotated or predicted function[18,20,21].

To date, the only known component of the jumbo phage nuclear shell is ChmA[6,7,10]. Among our list of RecA-interacting and ChmA-interacting proteins, we identified six proteins (gp2, gp61, gp63, gp64, gp148 and gp375) that clearly localized to the nuclear shell upon ΦPA3 infection of *P. aeruginosa* cells (Fig. 1d and Supplementary Table 3).

**Table 2 | Top 25 identified proteins from ΦPA3 RecA (gp175) miniTurboID**

| Protein | NCBI accession number | Times detected in three trials | log$_2$ (fold change)[a] | Normalized peak area[b] | Annotation | Found in ChmA miniTurbo ID? |
|---|---|---|---|---|---|---|
| gp175 (RecA) | YP_009217254.1 | 3 | 6.99 | 1.09×10$^{-2}$ | UvsX protein (RecA) | |
| gp210 | YP_009217289.1 | 3 | 8.38 | 2.96×10$^{-3}$ | Putative endonuclease | |
| gp253 | YP_009217332.1 | 1 | 7.47 | 1.59×10$^{-3}$ | Hypothetical protein | |
| gp222 | YP_009217301.1 | 1 | ∞ | 1.46×10$^{-3}$ | Hypothetical protein | Yes |
| gp313 | YP_009217392.1 | 2 | 3.31 | 7.11×10$^{-4}$ | Hypothetical protein | |
| gp49 | YP_009217132.1 | 1 | 3.73 | 6.16×10$^{-4}$ | Hypothetical protein | |
| gp358 | YP_009217437.1 | 1 | 8.02 | 4.91×10$^{-4}$ | Hypothetical protein | Yes |
| gp166 | YP_009217245.1 | 2 | ∞ | 2.12×10$^{-4}$ | Tail protein | Yes |
| gp78 | YP_009217158.1 | 2 | ∞ | 1.71×10$^{-4}$ | Endonuclease | Yes |
| gp271 | YP_009217350.1 | 1 | 5.18 | 1.41×10$^{-4}$ | Hypothetical protein | Yes |
| gp12 | YP_009217094.1 | 1 | ∞ | 1.27×10$^{-4}$ | Tail protein | Yes |
| gp200 | YP_009217279.1 | 1 | ∞ | 1.19×10$^{-4}$ | Hypothetical protein | Yes |
| gp233 | YP_009217312.1 | 1 | 6.33 | 9.00×10$^{-5}$ | SNF2 domain helicase | Yes |
| gp131 | YP_009217211.1 | 1 | ∞ | 7.80×10$^{-5}$ | Putative helicase | Yes |
| gp308 | YP_009217387.1 | 1 | ∞ | 7.22×10$^{-5}$ | Hypothetical protein | Yes |
| gp64 | YP_009217147.1 | 1 | ∞ | 7.05×10$^{-5}$ | Hypothetical protein | Yes |
| gp257 | YP_009217336.1 | 1 | 4.06 | 7.04×10$^{-5}$ | DNA ligase | Yes |
| gp144 | YP_009217223.1 | 1 | ∞ | 6.75×10$^{-5}$ | Hypothetical protein | |
| gp350 | YP_009217429.1 | 1 | ∞ | 5.58×10$^{-5}$ | Hypothetical protein | |
| gp239 | YP_009217318.1 | 1 | ∞ | 4.77×10$^{-5}$ | Hypothetical protein | Yes |
| gp108 | YP_009217188.1 | 1 | ∞ | 4.48×10$^{-5}$ | Hypothetical protein | |
| gp217 | YP_009217296.1 | 1 | ∞ | 3.10×10$^{-5}$ | Hypothetical protein | Yes |
| gp61 | YP_009217144.1 | 1 | ∞ | 2.42×10$^{-5}$ | Hypothetical protein | Yes |
| gp247 | YP_009217326.1 | 1 | ∞ | 1.76×10$^{-5}$ | Hypothetical protein | Yes |
| gp375 | YP_009217454.1 | 1 | ∞ | 1.70×10$^{-5}$ | Hypothetical protein | |
| gp62 | YP_009217145.1 | 1 | ∞ | 1.69×10$^{-5}$ | nvRNAP subunit | Yes |

[a]Fold change calculated as the fold change in average normalized peak area when comparing three trials with three negative-control trials. [b]Normalized peak area calculated as the fraction of the total peak area for ΦPA3 proteins (per dataset) assigned to a given protein.

One of these proteins, gp148, is predicted to be the portal protein of the phage capsid[18]. In other phages, the portal protein forms a homododecameric complex that orchestrates capsid assembly[22] and associates with the terminase to translocate genomic DNA into the capsid[23–27]. We previously showed that in this family of jumbo phages, capsids are docked on the nuclear shell for genomic DNA packaging[6], and our finding that the putative ΦPA3 portal protein associates with the shell suggests that it is directly responsible for capsid docking. Notably, we found that overexpressed GFP-tagged gp148 localized to the phage nuclear shell as early as 30 min postinfection, well before capsid assembly, and docking began at around 45 min postinfection[6] (Extended Data Fig. 3). This finding suggests that the portal protein can localize to the nuclear shell on its own, supporting the idea that it directly mediates capsid docking.

Apart from the portal protein, the remaining five nuclear-shell-localized proteins had no predicted function and were not found in previous mass spectrometry studies of mature jumbo phage virions[20]. Three of these five proteins (gp61, gp63 and gp64) were in a block of genes that is well-conserved across jumbo phages, whereas homologs of the remaining two proteins (gp2 and gp375) could be identified only in *Pseudomonas*-infecting jumbo phages (Extended Data Fig. 4). None of these five proteins shows detectable sequence homology to any other known protein. Moreover, all five proteins are expressed early in infections, with timing similar to the major nuclear shell protein ChmA[6,7]. Thus, we speculate that some or all of these proteins are components of the nuclear shell itself, and that they may mediate translocation of mRNA and/or proteins through the nuclear shell and mediate docking of capsids to the shell for genome packaging.

### gp2 is an interaction hub at the nuclear shell

Among the identified nuclear-shell-associated proteins, gp2 was among the most highly biotinylated proteins in our miniTurboID-ChmA samples (Table 1). We verified that GFP fusions of both ΦPA3 gp2 and its homolog from the related jumbo phage 201Φ2-1 (also gp2) colocalized with mCherry-fused ChmA in infected cells (Fig. 2a,b). To further define the interaction network of ΦPA3 gp2, we expressed GFP-fused gp2 in ΦPA3-infected *P. aeruginosa* cells, then purified the protein and interacting partners using GFP affinity chromatography. After purification of gp2 with a carboxy-terminal GFP tag, we identified two strong bands at a molecular weight of ~70 kDa on a silver-stained sodium dodecyl sulfate–polyacrylamide gel electrophoresis (SDS–PAGE) gel (Fig. 2c). We extracted a gel slice containing these two closely spaced bands and used trypsin mass spectrometry to identify the proteins. In this sample, the strongest signal (149 peptides, 81% sequence coverage) was for ChmA (gp53, 66.7 kDa), and the second strongest (41 peptides, 40% sequence coverage) was for the major capsid protein (gp136, 82.8 kDa; Supplementary Table 4). These data suggest that the observed doublet at ~70 kDa represents these two proteins and that gp2 interacts strongly with both ChmA and capsids.

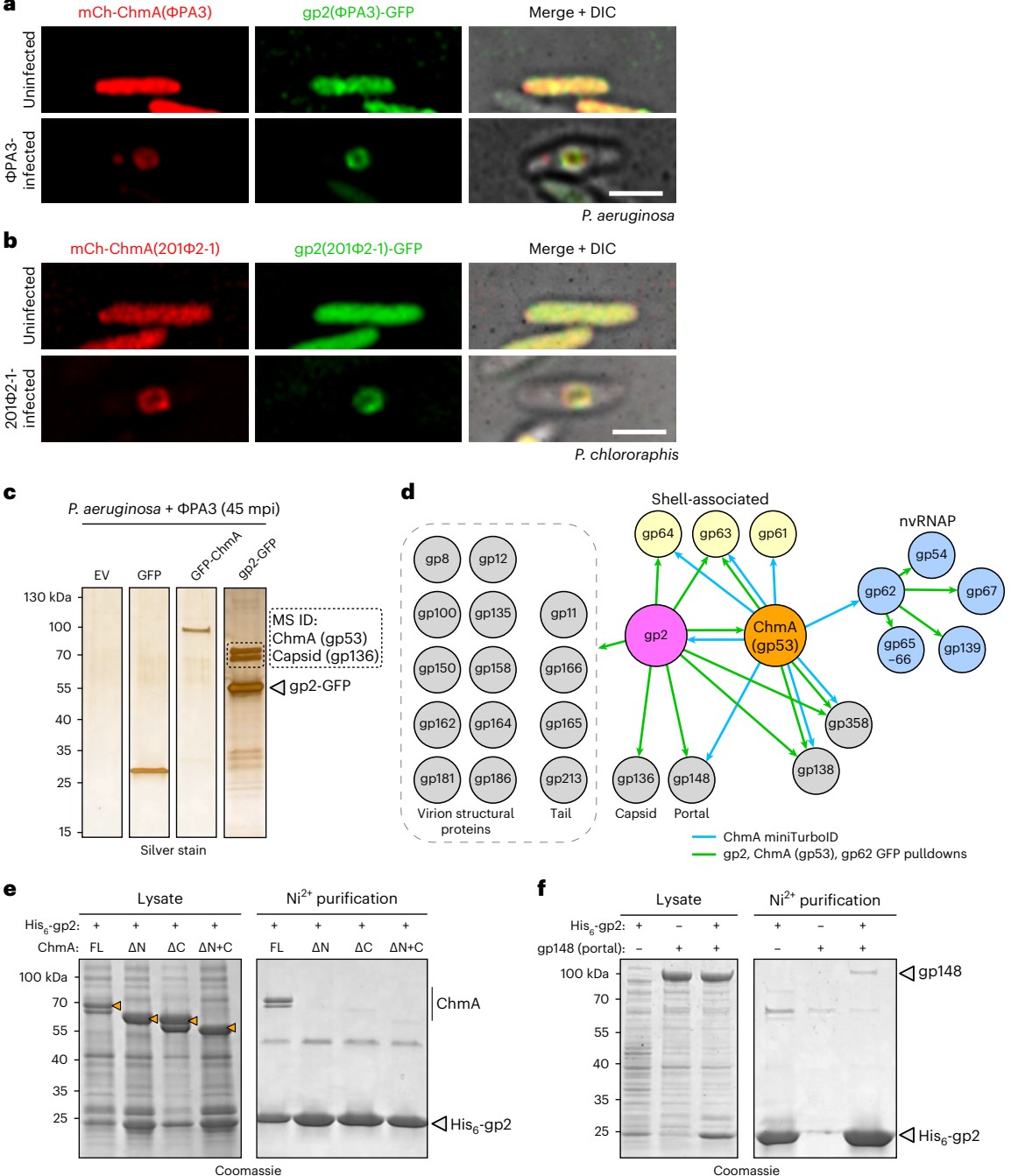

**Fig. 2 | gp2 is an interaction hub in the jumbo phage nuclear shell. a**, Colocalization of mCherry-fused ΦPA3 ChmA (red) and GFP-fused ΦPA3 gp2 (green) in *P. aeruginosa* cells. Scale bar, 2 μm. **b**, Colocalization of mCherry-fused 201Φ2-1 ChmA (gp105; red) and GFP-fused 201Φ2-1 gp2 (green) in *P. chlororaphis* cells. Scale bar, 2 μm. **c**, Silver-stained SDS–PAGE analysis of GFP pulldown experiments. EV, empty vector. Dotted box indicates the gel slice that was cut out (of the same bands in a Coomassie blue-stained gel) for tryptic mass spectrometry protein identification (Supplementary Table 4). **d**, Interaction network of the jumbo phage nuclear shell, with blue arrows indicating interactions identified by ChmA miniTurboID and green arrows indicating

interactions identified in GFP pulldowns (see Extended Data Fig. 4a for SDS–PAGE gels of all analyzed GFP pulldown samples and Supplementary Table 5 for full data). **e**, Ni²⁺ pulldown analysis of *E. coli*-coexpressed 201Φ2-1 gp2 (His₆-tagged) and ChmA (full-length or truncated: ΔN missing residues 1–63, ΔC missing residues 583–631 and ΔN + C missing residues 1–63 and 583–631). Doublet bands for ChmA arise from a methionine codon at position 33 of the annotated gene. Orange marks show the presence of ChmA in the lysates. See Extended Data Fig. 4b for control pulldown. **f**, Ni²⁺ pulldown analysis of *E. coli*-coexpressed ΦPA3 gp2 (His₆-tagged) and portal (gp148). DIC, differential interference contrast microscopy.

To further investigate the interactions between nuclear-shell-associated proteins, we next performed mass spectrometry on the full purified samples from GFP-tagged ChmA, gp2 (both N-terminal and C-terminal GFP tags) and the putative nvRNAP subunit gp62 (Fig. 2d, Extended Data Fig. 5a and Supplementary Table 5). We confirmed that

this approach could successfully purify functional protein complexes, as we successfully identified all five subunits of the nvRNAP complex in the gp62-GFP pulldown[19] (Fig. 2d and Supplementary Table 5). The most enriched protein in GFP-tagged gp2 samples was ChmA, and we also identified the major capsid protein (gp136), the predicted

portal protein (gp148), and 14 other proteins annotated as either tail or virion structural proteins (Fig. 2d and Supplementary Table 5). Two predicted phage tail proteins (gp213 and gp11) were also identified in pulldowns with gp62, suggesting that these proteins may simply be highly abundant in cell lysates (Supplementary Table 5). Nonetheless, the strong enrichment of phage structural proteins in GFP-tagged gp2 pulldowns strongly suggests that gp2 interacts directly with capsids, potentially as they dock on the nuclear shell for genomic DNA packaging. Also identified in the GFP-tagged gp2 pulldowns were two other shell-associated proteins, gp63 and gp64 (Fig. 2d and Supplementary Table 5), suggesting that these proteins may interact either directly with gp2 or indirectly through the ChmA lattice. Finally, two additional proteins (gp138 and gp358) were identified in both the ChmA and gp2 GFP pulldowns, as well as having been detected by ChmA miniTurboID labeling (Fig. 2d and Supplementary Tables 1 and 5). These two proteins are conserved across jumbo phages infecting *Pseudomonas* but have no annotated or predicted function, leaving their potential roles unknown.

## gp2 interacts with ChmA and the phage portal protein

We and others have shown that the nuclear shell in diverse jumbo phages including ΦPA3 is composed primarily of ChmA[5–7], which self-assembles into closed structures *in vitro* and forms a flexible lattice that surrounds the phage genome in phage-infected cells[10,11,28]. To confirm that gp2 interacts directly with ChmA, we coexpressed 201Φ2-1 gp2 and ChmA in *Escherichia coli* and performed Ni$^{2+}$ pulldowns using a His$_6$-tag on gp2 (ΦPA3 ChmA is poorly expressed in *E. coli*, precluding analysis in this phage). We found that His$_6$-tagged 201Φ2-1 gp2 robustly interacted with ChmA in this assay, showing that the two proteins directly interact (Fig. 2e and Extended Data Fig. 5b). We next deleted the N- and C-terminal segments of ChmA (NTS and CTS, respectively), which bind to neighboring protomers in the ChmA lattice to mediate nuclear shell assembly. In vitro, deletion of either NTS or CTS led to loss of ChmA self-assembly[10], and we found that gp2 was unable to interact with ChmA mutants lacking NTS, CTS, or both NTS and CTS (Fig. 2e). The loss of gp2 binding when deleting either the ChmA NTS or CTS suggests that gp2 does not simply bind one of these tail segments; rather, the data suggest that gp2 interacts specifically with the assembled ChmA lattice.

We next performed a similar coexpression experiment with His$_6$-tagged ΦPA3 gp2 and the portal protein gp148. We found that gp2 could interact directly with gp148 in this assay (Fig. 2f). Combined with our data showing that gp2 interacts directly with self-assembled ChmA, these data suggest that gp2 is an integral component of the phage nuclear shell that is directly involved in the docking and filling of capsids through an interaction with the portal. Based on its localization and probable crucial role in phage nuclear structure and function, we name ΦPA3 gp2 and its homologs in related jumbo phages chimallin B (ChmB).

## ChmB forms a homodimer with a novel fold

To determine the structural basis for ChmB interactions with other proteins, we recombinantly purified the protein from several jumbo phages and determined a 2.6 Å resolution crystal structure of ChmB from the related phage PA1C (gp2; 38% identical to ΦPA3 gp2) (Table 3). ChmB formed a homodimer in solution (Fig. 3a and Extended Data Fig. 6a–c), and the structure revealed an intertwined dimeric structure with the N terminus of each protomer forming a short β-strand and an α-helix that pack against the C-terminal globular domain of its dimer mate. Overall, the ChmB dimer adopts a distinctive U shape with dimensions of ~5 × 8 nm (Fig. 3b). Searches with the DALI or FoldSeek protein structure comparison tools[29,30] showed no known structural relatives.

## ChmB point mutants disrupt phage nucleus formation

To determine the roles of ChmB in phage nucleus formation and function, we first overexpressed GFP-tagged wild-type ΦPA3 ChmB

**Table 3 | Data collection and refinement statistics**

| | PA1C gp2 |
|---|---|
| **Data collection** | |
| Space group | P2$_1$ |
| Cell dimensions | |
| $a, b, c$ (Å) | 60.96, 86.56, 71.88 |
| $α, β, γ$ (°) | 90, 110.94, 90 |
| Resolution (Å) | 67.13–2.63 (2.74–2.63)$^a$ |
| $R_{sym}$ or $R_{merge}$ | 0.184 (1.101) |
| $I/σ(I)$ | 4.8 (0.9) |
| Completeness (%) | 98.5 (94.0) |
| Redundancy | 3.3 (3.4) |
| **Refinement** | |
| Resolution (Å) | 67.13–2.63 |
| No. reflections | 20,421 |
| $R_{work}$ / $R_{free}$ | 27.10% / 30.01% |
| No. atoms | |
| Protein | 10,615 |
| Ligand/ion | 0 |
| Water | 0 |
| B factors | |
| Protein | 73.55 |
| Ligand/ion | NA |
| Water | NA |
| R.m.s. deviations | |
| Bond lengths (Å) | 0.0029 |
| Bond angles (°) | 0.55 |

$^a$Values in parentheses are for the highest-resolution shell. NA, not applicable.

in ΦPA3-infected *P. aeruginosa* cells (Fig. 4a). In uninfected cells, gp2 overexpression did not cause a significant growth defect, indicating that the protein is not inherently toxic (Extended Data Fig. 7a,b). In infected cells, we observed a striking increase in the average size of the phage nucleus and a concomitant increase in the nuclear DNA content (as measured by total DAPI signal within the nucleus) (Fig. 4b,c and Extended Data Fig. 8a–c). Further, a significant fraction of cells (42%) showed ChmB localization suggestive of multiple juxtaposed phage nuclei or a single phage nucleus with an aberrant shell structure (Fig. 4d).

We next aligned ChmB homologs from jumbo phages that infect *Pseudomonas* species (Extended Data Fig. 5d) and identified two highly conserved surface residues: Q53 and A159 (ΦPA3 gp2 numbering). We generated point mutations of these residues (Q53A and A159D, respectively) designed to alter the ChmB surface and potentially disrupt specific protein–protein interactions. Whereas ΦPA3-infected *P. aeruginosa* cells overexpressing wild-type ChmB showed large and/or multiple phage nuclei, infected cells overexpressing either ChmB-Q53A or A159D instead showed a strong disruption in nucleus formation and growth in a large fraction of cells. In around 20% of cells (23% for ChmB-Q53A, 17% for ChmB-A159D), the phage nuclear DNA signal in late infections resembled the puncta usually observed in very early infections, before significant phage nuclear DNA replication and nucleus growth[6] (Fig. 4e and Extended Data Fig. 8d,e). In another population of cells (5% for ChmB-Q53A, 7% for ChmB-A159D), nuclear DNA appeared to be entirely absent despite these cells sometimes showing ChmB localization reminiscent of a phage nuclear shell (Fig. 4e). Finally, a third

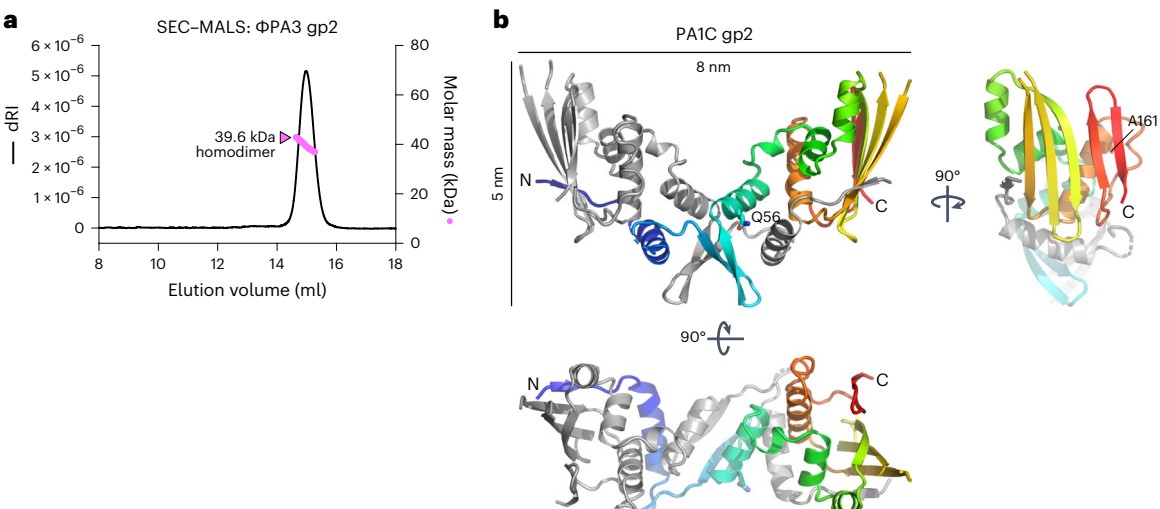

**Fig. 3 | Structure of gp2. a**, SEC–MALS of ΦPA3 gp2, showing that it is homodimeric in solution (monomer molecular weight = 22.5 kDa). See Extended Data Fig. 5a–c for SEC–MALS analysis of other jumbo phage gp2 proteins. **b**,

Structure of the PA1C gp2 homodimer, with one protomer colored gray and the other colored as a rainbow from N terminus (blue) to C terminus (red). dRI, differential refractive index protein concentration measurement.

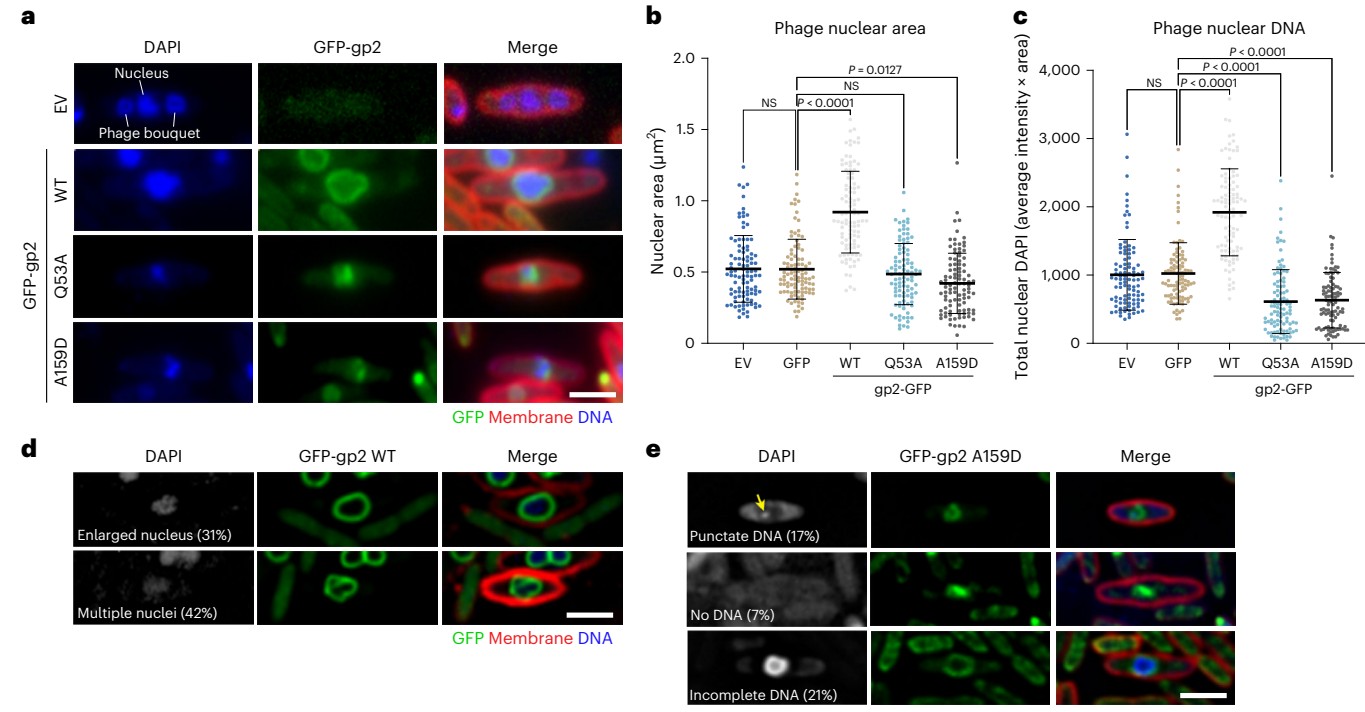

**Fig. 4 | gp2 mutations cause defects in phage nucleus formation and morphology. a**, Fluorescence imaging of ΦPA3-infected *P. aeruginosa* cells expressing no additional proteins, GFP-tagged wild-type gp2 or GFP-tagged gp2 point mutants. Undeconvolved images that were used for DAPI quantitation (**b** and **c**) are shown. GFP is shown in green, FM4-64 (to visualize membranes) in red and DAPI (to visualize nucleic acids) in blue. Scale bar, 2 μm. **b**, Phage nuclear area of ΦPA3-infected *P. aeruginosa* cells expressing no additional proteins, GFP-tagged wild-type gp2 or GFP-tagged gp2 point mutants. *n* = 100 for all samples; error bars represent mean ± s.d. *P* values were calculated from one-way analysis of variance (ANOVA) tests. Errors bars show mean ± s.d. **c**, Total nuclear DNA in

ΦPA3-infected *P. aeruginosa* cells expressing no additional proteins, GFP-tagged wild-type gp2 or GFP-tagged gp2 point mutants, calculated by multiplying each cell's average DAPI signal within the nucleus by that cell's nuclear area (**b**). *P* values were calculated from one-way ANOVA tests. Errors bars show mean ± s.d. **d**, Visual phenotypes observed in ΦPA3-infected *P. aeruginosa* cells expressing GFP-tagged wild-type gp2 (*n* = 100 cells). See Extended Data Fig. 6c for additional examples. Scale bar, 2 μm. **e**, Visual phenotypes observed in ΦPA3-infected *P. aeruginosa* cells expressing GFP-tagged gp2 A159D (*n* = 100 cells). See Extended Data Fig. 6d for additional examples and Extended Data Fig. 6e for examples of similar phenotypes from gp2 Q53A. Scale bar, 2 μm. NS, not significant; WT, wild type.

population of cells (14% for ChmB-Q53A, 21% for ChmB-A159D) showed an aberrant nuclear shell structure with nuclear DNA staining that appeared to incompletely fill the nuclear area (Fig. 4e). Overall, nearly half of infected cells expressing mutant ChmB (42% for ChmB-Q53A,

45% for ChmB-A159D) showed abnormal nuclear shell and/or nuclear DNA morphology.

To test whether the ChmB point mutants affected oligomerization or binding to known partner proteins, we first purified

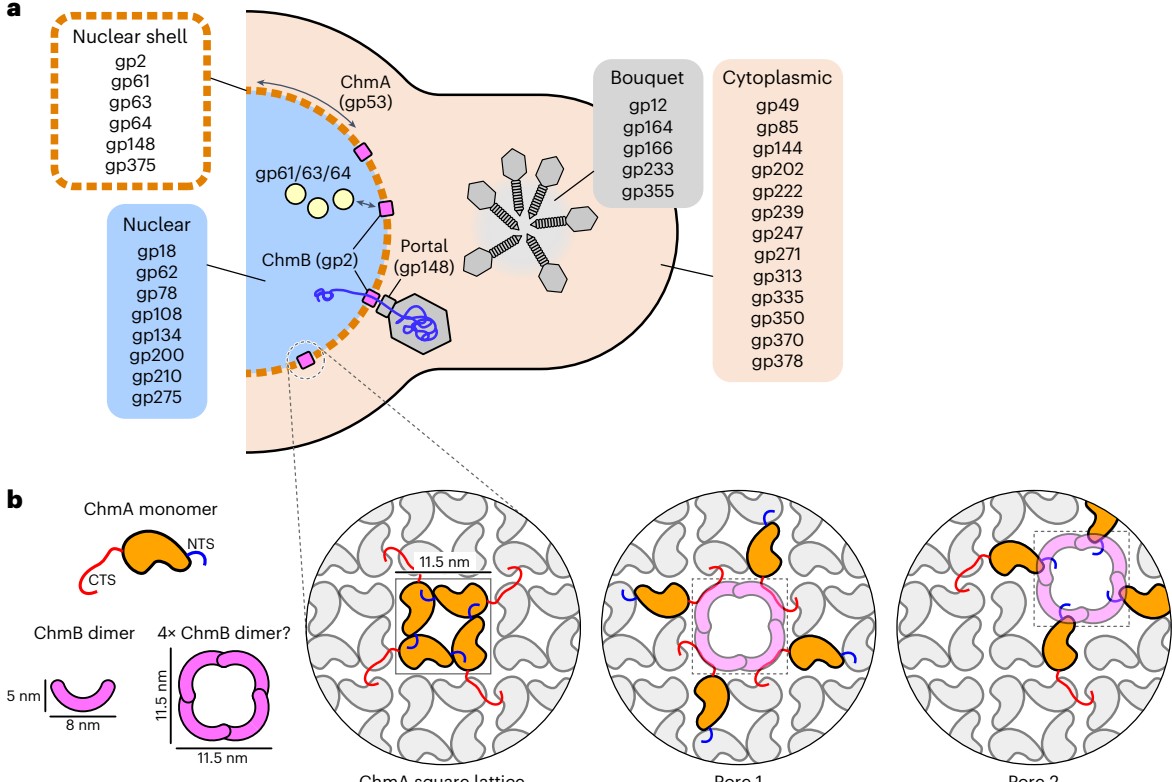

**Fig. 5 | Model for jumbo phage protein localization and nuclear shell architecture and function. a**, Schematic of a ΦPA3-infected *P. aeruginosa* cell with assembled phage nucleus (blue) bounded by a ChmA lattice (orange). Proteins that we found to localize to the nucleus, nuclear shell, phage bouquet or cytoplasm are listed. ChmB (pink) is shown integrated into the ChmA lattice, where it may mediate the docking of phage capsids by binding the portal protein, for genomic packaging. Further interactions with gp61, gp63 and/or gp64 (light yellow) or other shell-associated proteins could accommodate mRNA export or specific protein import. **b**, Schematic of the ChmA lattice derived from cryoelectron tomography analysis of intact 201Φ2-1 and Goslar nuclear shells. ChmA is shown in orange, with N-terminal and C-terminal tails shown in blue and red, respectively. Removal of four contiguous ChmA protomers from the lattice would leave a cavity of ~11.5 × 11.5 nm (two possibilities shown), which could be filled by an assembly of ChmB to generate a pore.

wild-type and mutant ΦPA3 ChmB and analyzed the purified proteins by size-exclusion chromatography. All three proteins showed similar elution profiles, indicating that ChmB oligomerization was not affected by the mutations (Extended Data Fig. 7c). Next, we coexpressed His6-tagged 201Φ2-1 ChmB (wild-type and mutants equivalent to ΦPA3 ChmB-Q53A and A159D) with untagged ChmA. Neither mutation affected ChmA association in our pulldown assay (Extended Data Fig. 7d), consistent with our observation that ChmB point mutants localized properly to the phage nuclear shell in infected cells (Fig. 4a). Similarly, neither ChmB mutation affected the ability of His6-tagged ΦPA3 ChmB to associate with the portal protein (gp148) in a coexpression assay (Extended Data Fig. 7e). Thus, the observed effects of ChmB point mutants on phage nuclear shell development are probably not caused by a failure to interact with ChmA or the phage capsids. Rather, these effects may involve other putative interaction partners of ChmB, such as the uncharacterized nuclear-shell-associated proteins gp61, gp63 and gp64.

## Discussion

In contrast to well-studied small-genome phages, nucleus-forming jumbo phages build several characteristic structures in infected cells, including the phage nucleus[6,7,9,15], PhuZ spindle[6,7,13,31] and phage bouquets[14,15]. Here, we combined proximity labeling with subcellular localization analysis by fluorescence microscopy to establish a subcellular protein localization map comprising 44 phage-encoded proteins (Fig. 5a). Although this map is incomplete, it nonetheless represents a major step in our functional understanding of this distinctive family of phages.

Here, we focus on the phage nucleus, which separates the replicating phage genome from the host cytoplasm and protects the phage from DNA-targeting immune factors encoded by the host. We and others have shown that the jumbo phage nuclear shell is predominantly composed of a single layer of the phage-encoded protein chimallin (ChmA), which assembles into a lattice with pores less than 2 nm in width[10,11]. As these pores are likely to be too small for the passage of nucleic acids or proteins, we theorized that the ChmA lattice incorporates additional components that mediate mRNA and protein translocation through the phage nuclear shell. Further, based on our previous observation that capsids dock on the nuclear shell for genome packaging[5], this structure must also incorporate components that mediate capsid docking and enable the passage of genomic DNA through the nuclear shell.

Our proximity labeling and localization mapping approach identified six proteins in the jumbo phage ΦPA3 that associate with the phage nuclear shell. We found that one of these proteins (gp2) associated directly with self-assembled ChmA in vitro, suggesting that it is an integral phage nuclear shell protein and prompting us to name it ChmB. ChmA self-assembles into a flexible square lattice with extended NTS and CTS binding to neighboring protomers[10,11] (Fig. 5b). NTS-mediated interactions define ChmA homotetramers that measure ~11.5 × 11.5 nm, whereas CTS-mediated interactions mediate interactions between neighboring tetramers. The distinctive U shape and overall dimensions of the ChmB dimer, at ~5 × 8 nm, suggest a model in which multiple ChmB dimers could line a hole created by the removal of four or more neighboring ChmA protomers from the lattice (two possibilities shown in Fig. 5b). Indeed, size-exclusion chromatography of ChmB at high

concentration suggests a propensity for higher-order self-assembly that may be reinforced by integration into the ChmA lattice (Extended Data Fig. 7c).

In addition to associating directly with ChmA, ChmB is at the center of a large protein–protein interaction network in phage-infected cells. ChmB interacts directly with the phage portal protein, and the portal protein can also localize on its own to the phage nuclear shell in infected cells. Based on these data, we propose that ChmB mediates capsid docking and genome packaging at the nuclear shell through a direct interaction with the portal protein. ChmB may also interact with other nuclear-shell-associated proteins such as gp61, gp63 and gp64 to mediate specific translocation of mRNA and protein through the shell.

Overexpression of wild-type ChmB in infected cells resulted in phage nuclei that were significantly larger than normal, often showing aberrant morphology and appearing as multiple distinct phage nuclei in a single cell. Conversely, overexpression of mutant ChmB proteins (Q53A or A159D) led to the formation of phage nuclei that were significantly smaller or appeared to be completely devoid of DNA based on DAPI staining. Importantly, these point mutations did not disrupt ChmA binding or ChmB localization to the phage nuclear shell, nor did they disrupt binding to the phage portal protein. The strong phenotypic effects of ChmB point-mutant expression in infected cells therefore suggest that ChmB has additional binding partners with fundamental roles in the formation and maturation of the phage nucleus.

ChmB is conserved among all known nucleus-forming jumbo phages that infect *Pseudomonas*, but it is not found in distantly related phage such as *E. coli* phage Goslar[15] or *Serratia* phage PCH45 (ref. 9) (Extended Data Fig. 4). Given its apparent role in phage nuclear shell function, why is ChmB not more widely conserved? The most likely explanation is that it is conserved, but that distant homologs of ChmB are too divergent to be recognized by sequence-based searches. This scenario is supported by the high sequence divergence of gp2 homologs relative to other important proteins such as ChmA: across five representative jumbo phages infecting *Pseudomonas* (ΦPA3, PA1C, Phabio, 201Φ2-1 and ΦKZ), pairwise sequence identities for ChmA homologs average 51%, whereas ChmB proteins average 27% identity (Extended Data Fig. 4). Alternatively, a different protein might perform a role equivalent to that of ChmB in other nucleus-forming jumbo phage families.

In summary, this work provides new insights into the organizational principles of nucleus-forming jumbo phages and the molecular mechanisms of the phage nucleus in particular. Our results identify a protein interaction network centered around the phage nucleus that will form the basis for future research in this area. We identified a key protein, ChmB, at the center of this network that is likely to have multiple roles both as a pore for macromolecular translocation through the nuclear shell and for capsid docking and genomic DNA packaging. Further work will be required to fully understand the composition of the phage nucleus and the myriad proteins that contribute to its remarkable functions.

## Online content

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

## Methods

### Strains, growth condition and phage preparation

*Pseudomonas chlororaphis* strain 200-B cells were cultured on solid hard agar (HA). *P. aeruginosa* strains PA01 and K2733 (efflux pump knockout; *ΔMexAB-OprMΔMexCD-OprJΔMexEF-OprNΔMexXY-OprM*)[32] were cultured in Luria-Bertani (LB) media. For amplification of phages, host strains were cultured in liquid media at 30 °C overnight; then, 20 μl of high-titer phage lysate was mixed with 100 μl of cells with optical density at wavelength 600 nm ($OD_{600}$) of 0.6 (*P. chlororaphis* for 201Φ2-1, *P. aeruginosa* for ΦPA3), incubated for 20 min at room temperature, mixed with 5 ml HA (for 201Φ2-1 in *P. chlororaphis*) or LB top agar (for phage ΦPA3 in *P. aeruginosa*) and poured over an HA or LB plate. The plates were then incubated upside-down at 30 °C overnight. The next day, the plates were incubated with 5 ml of phage buffer for 5 h at room temperature. Lysates were collected and centrifuged at 21,000$g$ for 10 min. Supernatants were stored at 4 °C with 0.01% chloroform.

### Plasmid construction and bacterial transformation

Genes of interest were PCR-amplified from high-titer phage lysates then ligated into linearized plasmid backbones using an NEBuilder HiFi DNA Assembly Cloning Kit (New England Biolabs, catalog number E5520S). For protein expression in *P. chlororaphis* and *P. aeruginosa*, the pHERD30T vector was used[33]. For overexpression in *E. coli*, UC Berkeley MacroLab vectors 2-BT (ampicillin resistant, His$_6$-TEV tag; Addgene, catalog number 29666) and 13-S (spectinomycin resistant, no tag; Addgene, catalog number 48323) were used. Recombinant plasmids were transformed into *E. coli* DH5α and plated on LB agar containing appropriate antibiotics (25 μg ml$^{-1}$ gentamicin sulfate, 100 μg ml$^{-1}$ ampicillin or 100 μg ml$^{-1}$ spectinomycin as appropriate). Constructs were confirmed by DNA sequencing and subsequently introduced into indicated organisms of interest and selected on LB supplemented with antibiotics. Selected overnight cultures were stored in 25% glycerol at −80 °C. See Supplementary Table 6 for all plasmids used in this study and Supplementary Table 7 for sequences of all oligonucleotides and synthesized genes.

### Fluorescence microscopy of single-cell-infection assay

Agarose pads (1.2%) were prepared on concavity slides. Each pad was supplemented with 0.05–1.00% arabinose to induce protein expression. In certain experiments, FM4-64 (1 μg ml$^{-1}$) was added to stain cell membranes, and DAPI (1 μg ml$^{-1}$) was added to stain the nucleoid. *P. chlororaphis* and *P. aeruginosa* strains were inoculated on the pads and grown in a humid chamber at 30 °C for 2 h. For phage infections, 10 μl of phages (10$^8$ plaque-forming units (PFU) ml$^{-1}$) were added on the cells, followed by incubation for an additional 40 min at 30 °C to allow infection to proceed. At the desired time points, the pads were sealed with a coverslip, and fluorescence microscopy was performed with a DeltaVision Spectris Deconvolution Microscope (Applied Precision). Cells were imaged using at least eight images in the $z$ axis from the middle focal plane in 0.15 μm increments. For time-lapse imaging, 8 Z points were selected and subsequently imaged using the UltimateFocus mode. Images were further processed using a deconvolution algorithm (DeltaVision SoftWoRx Image Analysis Program) and analyzed in Fiji[34].

### Phage nucleus area and DAPI measurements from fixed cells

Agarose pads (1.2%) were prepared on concavity slides containing 1% arabinose plus 1 μg ml$^{-1}$ FM4-64. *P. aeruginosa* strains were inoculated on pads and grown for 2 h in a humid chamber at 30 °C. Seventy-five minutes postinfection with 10 μl of phage (10$^8$ PFU ml$^{-1}$), 20 μl of fixation mixture (1.5% glutaraldehyde, 13.76% paraformaldehyde and 0.16 M NaPO$_4$ at pH = 7.4) was added to each pad, followed by incubation at room temperature for 20 min. For high levels of nucleoid staining for quantifications, 20 μl of 20 μg ml$^{-1}$ DAPI was added to the fixed cells, followed by incubation for 20 min at room temperature. Pads were sealed with coverslips, and fluorescence microscopy was performed as above. For quantitation of DAPI area and intensity in nucleoids or phage nuclei, images before

deconvolution were analyzed using Fiji with built-in measurement tools. Statistical analysis was performed in GraphPad Prism.

### Proximity labeling with miniTurboID

For proximity labeling, overnight cultures of *P. aeruginosa* were grown in LB media with 25 μg ml$^{-1}$ gentamicin sulfate. Cultures were then diluted to $OD_{600}$ = 0.1 and supplemented with 500 μM biotin. When cells reached $OD_{600}$ = 0.5, they were diluted 1:10 in 50 ml total volume in 250 ml flasks and grown in LB supplemented with 0.1% arabinose, 500 μM biotin, 25 μg ml$^{-1}$ gentamicin sulfate and 0.2 mM CaCl$_2$. When the cells reached $OD_{600}$ = 0.3, they were infected with phage ΦPA3 at a multiplicity of infection of 3. At 45 min postinfection, cultures were collected and centrifuged at 3,000$g$ at 4 °C. Cell pellets were stored at −80 °C for mass spectrometry.

### Mass spectrometry

To prepare biotinylated samples for immunoprecipitation and mass spectrometry, frozen cell pellets (100 μl) were thawed and resuspended in 100 μl water. Ten microliters of resuspended cells were mixed with 200 μl of 6 M guanidine-HCl, vortexed and then subjected to three cycles of incubation at 100 °C for 5 min, followed by cooling to room temperature. Then, 1.8 ml of pure methanol was added to the boiled cell lysate, followed by vortexing, incubation at −20 °C for 20 min, and centrifugation at 21,000$g$ for 10 min at 4 °C. The tube was inverted and dried to remove any liquid, and the pellet was resuspended in 200 μl of 8 M urea in 0.2 M ammonium bicarbonate. The mixture was incubated for 1 h at 37 °C with constant agitation. Following the incubation, 4 μl of 500 mM Tris(2-carboxyethyl) phosphine and 20 μl of 400 mM chloro-acetamide were added. Protein concentration was measured by BCA assay from a 10 μl sample; then, 600 μl of 200 mM ammonium bicarbonate was added to bring the urea concentration to 2 M. One microgram of sequencing-grade trypsin was added for each 100 μg of protein in the sample, followed by incubation overnight at 42 °C. Following trypsin incubation, 50 μl of 50% formic acid was added (ensuring that the pH dropped to 2 using pH test strips), and then samples were desalted using C18 solid phase extraction (Waters Sep-Pak C18 12 cc Vac Cartridge, catalog number WAT036915) as described by the manufacturer protocol. The peptide concentration of each sample was measured using BCA after resuspension in 1 ml phosphate-buffered saline (PBS) buffer.

For biotin immunoprecipitation, 200 μl of 50% slurry of NeutrAvidin beads (Pierce) was washed three times with PBS; then, 1 mg of resuspended peptide solution in PBS was added, followed by incubation for 1 h at room temperature. Beads were washed three times with 2 ml PBS plus 2.5% acetonitrile (ACN) and once in ultrapure water, and excess liquid was carefully removed with a micropipette. Biotinylated peptides were eluted twice with 300 μl of elution buffer (0.2% trifluoroacetic acid, 0.1% formic acid and 80% ACN in water), with the second elution involving two 5 min incubations at 100 °C. Samples were then dried completely before being subjected to mass spectrometry.

### Liquid chromatography coupled with tandem mass spectrometry

Trypsin-digested peptides were analyzed by ultra-high-pressure liquid chromatography coupled with tandem mass spectroscopy using nanospray ionization. The nanospray ionization experiments were performed using a Orbitrap Fusion Lumos hybrid mass spectrometer (Thermo) interfaced with nanoscale reverse-phase ultra-high-pressure liquid chromatography (Thermo Dionex UltiMate 3000 RSLC Nano System) using a 25 cm, 75 μm ID glass capillary packed with 1.7 μm C18 (130) BEH beads (Waters Corporation). Peptides were eluted from the C18 column into the mass spectrometer using a linear gradient (5–80%) of ACN at a flow rate of 375 μl min$^{-1}$ for 3 h. The buffers used to create the ACN gradient were as follows: buffer A (98% H$_2$O, 2% ACN, 0.1% formic acid); and buffer B (100% ACN, 0.1% formic acid). The mass spectrometer parameters were as follows: an MS1 survey scan using the Orbitrap

detector (mass range ($m/z$): 400–1,500 (using quadrupole isolation), 120,000 resolution setting, spray voltage of 2200 V, ion transfer tube temperature of 275 °C, AGC target of 400,000 and maximum injection time of 50 ms) was followed by data-dependent scans (top speed for the most intense ions, with charge state set to only include +2–5 ions and a 5 s exclusion time, selecting ions with minimal intensities of 50,000) in which the collision event was carried out in the high-energy collision cell (higher-energy collisional dissociation collision energy of 30%), and the fragment masses were analyzed in the ion trap mass analyzer (with an ion trap scan rate of turbo, the first mass $m/z$ was 100, the AGC target was 5,000 and the maximum injection time was 35 ms). Protein identification and label-free quantification were carried out using Peaks Studio 8.5 (Bioinformatics Solutions Inc.). Variable modification at lysine residues of +226.08 atomic mass units was used in the peptide sequencing parameters.

## Mass spectrometry analysis

For each sample, biotinylated peptides identified by mass spectrometry were divided into host (*P. aeruginosa*) and phage (ΦPA3) peptides. Host peptides that were identified in all samples were used to normalize phage peptide signals across the dataset. Biotinylated phage protein peak areas were calculated by summing the peak areas of each peptide assigned to a given protein. The fold changes for proteins from ChmA (gp53)-miniTurboID and RecA (gp175)-miniTurboID were calculated by comparison with protein peak areas from GFP-miniTurboID samples. The proteins were sorted according to their average normalized fold changes.

The genome sequence of ΦPA3 (NCBI RefSeq NC_028999.1) is misannotated between the coding regions for gp64 (NCBI accession YP_009217147.1, nucleotides 51942–53267) and gp68 (NCBI accession YP_009217148.1, nucleotides 58478–60010). We manually annotated this region to identify gp65–66 (which together code for a single protein, separated by an intron spanning nucleotides 55005–55454)[19] and gp67 for identification by mass spectrometry:

>gp65–66 (NC_028999.1 nucleotides 53811-55004 and 55455-56447)

MYEEHNLRRAVREIHAKLLGHAALDPYYGTTSAARGAM
FLSHIGQAPVVEGNEPRRVMTGMEMRYAEYTFDVRLPTDCTILHKVRKY
PTGQGYGAIQHNPVTTLIYENYYDEYKTIGVLHVPEYMSFHQDF
GYELVKNKEVWESLQPDQMFAKDTVIAQSSTVKSNGLYGMGVNAN
VAFMSVPGTIEDGFVVSDEFLERMSPRTYTTAVCGAGKKAFFLNMYGD
DKIYKPFPDIGEKIREDGVIFAVRDLDDDLAPAEMTPRALRTLDRTFDRAV
IGDPGATVKDIKVYWDERQNPSFTPSGMDGQLRKYYDALCTYYREIIKI
YRGLLARRKDKLRISEEFNQLLVEAMIYLPQAEGQRKLTRMYRLEQLDE
WRVELTYESIKVPGGAYKLTDFHGGKGVVCEVRPKADMPVDEFGNVVDAI
IFGGSTMRRSNYGRIYEHGFGAASRDLAQRLRVEAGLPRHGVVPEQDLN
RVCSNREWVTYAFAELQEFYYIIAPTMHEILREHPSPAEYVKTVLRDGFSYI
YSPVDDPVDLMSSLNCIMNSRFCPNHTRVTYRGQDGKMVTTKDKVLVG
PLYMMLLEKIGEDWSAVASVKVQQFGLPSKLNNSDRSSTPGRESAIRSFGE
SETRSYNCTVGPEATVELLDQTNNPRAHLAVINSILTADKPSNIERAVDRT
KVPFGSSRPVDLLEHLLECRGLKFEYATTDGVQPVHTAVPIRAQQKVK
SEAIEE*

>gp67 (NC_028999.1 nucleotides 56450-58420)

MNQYNARDLLNMSYDDLFAIPNEWHKIIFDDGEILTKDRATKLSILL
WHPLKQFPNATLSVKYHLGDTRVTSKSLVKLLNSVIWGIHAWSNEQVD
PEVLARLAIEAKNVLYNEATSRLGAYVATLSMFEIAEVYNHPKVREAN
QNIEPTTHGIETIAYGKIKEAFNDPTQFRGNSIIEGLRSGTQKMEQLLQAF
GPRGFPTDINSDIFAEPCLTGYIDGIWGLYENMIESRSGTKALLYNKELL
RVTEYFNRKSQLIAQYVQRLHKGDCGAGYIEFPVIKAYLKSLRGKFYL
NEETGKREILQGNETHLIGKKIKMRSVLGCVHPDPQGICATCYGTLAD
NIPRGTNIGQVSAVSMGDKITSSVLSTKHTDATSAVEQYKITGVEAKYL
REGQAPETLYLKKELANKGYRLMIGRNEAQNLADVLMIDNLSAYPPT
SASELTRIGLVRTVDGIDEGDVLTVSLYNRKASLSIELLQHVKRVRWELD
NRDNIVIDLNGFDFSLPFLTLPYKHVNMYEVMKRIQSFLHSGSDTEG
SKLSSDKVGFTSKTYLKNYNDPIDAVAAFASLVNEKIQLPMPHCEVLVY

AMMVRSTQQRDYRLPKPGISGQFEKYNKLMQSRSLAGAMAFEKQHE
PLNNPGSFLYTLRNDHPYDLAVKGGKLY*.

## GFP pulldowns

For GFP pulldowns, GFP-Trap Magnetic Agarose beads (Chromotek gtma-20) were used. Overnight cultures of *P. aeruginosa* expressing GFP-tagged proteins of interest were grown in LB plus 25 μg ml⁻¹ gentamicin sulfate. Cells were diluted to an $OD_{600}$ of 0.1 then grown further to an $OD_{600}$ of 0.5. Cultures were diluted 1:10 into 50 ml total volume of culture in 250 ml flasks and grown in LB supplemented with 0.1% arabinose, 25 μg ml⁻¹ gentamicin sulfate, and 0.2 mM $CaCl_2$. When the cells reached $OD_{600} = 0.3$, they were infected with ΦPA3 at a multiplicity of infection of 3. Cultures were collected at 45 min postinfection and centrifuged at 3,000 $g$ at 4 °C. Cell pellets were stored at −80 °C.

For the GFP pulldown, thawed cell pellets were incubated for 1 h with 500 μl lysis buffer (10% glycerol, 25 mM Tris (pH 7.5), 150 mM NaCl, 4 mg ml⁻¹ lysozyme, 20 μg ml⁻¹ DNase I, 2× cOmplete Protease Inhibitor, 0.4 mM phenylmethylsulfonyl fluoride). Cell suspensions were sonicated for 10 rounds with 20 pulses per round (Duty Cycle 40, Output 4). Lysed cells were centrifuged for 30 min at 21,000 $g$ at 4 °C. For each sample, 25 μl of bead slurry (prewashed into dilution buffer: 10 mM Tris-HCl pH 7.5, 150 mM NaCl, 0.5 mM EDTA) was used. Then, 500 μl of cell lysate was added to the beads, followed by rotation end-to-end for 1 h at 4 °C. Beads were washed five times with wash buffer (10 mM Tris/Cl pH 7.5, 150 mM NaCl, 0.05% NP-40 substitute, 0.5 mM EDTA). For SDS–PAGE, cells were resuspended with 2× SDS buffer (120 mM Tris-HCl pH 6.8, 20% glycerol, 4% SDS, 0.04% bromophenol blue, 10% β-mercaptoethanol) and boiled at 100 °C for 5 min. Samples (10 μl) of each elution were run on two separate SDS–PAGE gels and visualized by either silver staining or Coomassie blue staining. For tryptic mass spectrometry of gel bands, bands were cut out of Coomassie blue-stained gels. The remaining 80% of each elution was used for mass spectrometry identification of proteins as described above.

## Protein purification

Full-length phage gp2 protein sequences were codon-optimized and synthesized (Invitrogen/GeneArt), then cloned into UC Berkeley Macrolab vector 2-BT (Addgene, catalog number 29666) to generate constructs with N-terminal TEV protease-cleavable His₆-tags. Proteins were expressed in *E. coli* strain Rosetta 2 (DE3) pLysS (EMD Millipore). Cultures were grown at 37 °C to $OD_{600} = 0.7$ then induced with 0.25 mM IPTG and shifted to 20 °C. After a 16 h incubation, cells were harvested by centrifugation and resuspended in a buffer containing 25 mM Tris pH 7.5, 10% glycerol, 300 mM NaCl, 5 mM imidazole, 5 mM β-mercaptoethanol, and 1 mM $NaN_3$. Proteins were purified by Ni²⁺-affinity (Ni-NTA agarose, Qiagen) then passed over an anion-exchange column (HiTrap Q HP, Cytiva) in a buffer containing 100 mM NaCl, collecting flow-through fractions. Tags were cleaved with TEV protease[35], the mixture was passed over another Ni²⁺ column, and the flow-through fractions containing protease-cleaved proteins of interest were collected and concentrated. The protein was passed over a size-exclusion column (Superdex 200, Cytiva) in buffer GF (buffer A plus 300 mM NaCl and 1 mM dithiothreitol), then concentrated by ultrafiltration (Amicon Ultra, EMD Millipore) to 10 mg ml⁻¹ and stored at 4 °C.

For characterization of oligomeric state by size-exclusion chromatography coupled to multi-angle light scattering (SEC–MALS), 100 μl of purified proteins at 5 mg ml⁻¹ were injected onto a Superdex 200 Increase 10/300 GL column (Cytiva) in buffer GF. Light scattering and refractive index profiles were collected using miniDAWN TREOS and Optilab T-rEX detectors (Wyatt Technology), respectively, and molecular weight was calculated using ASTRA v.8 software (Wyatt Technology).

## Crystallization and structure determination

Purified PA1C gp2 in a buffer containing 20 mM Tris pH 8.5, 1 mM dithiothreitol, and 100 mM NaCl (14 mg ml⁻¹) was mixed 1:1 with well solution

containing 0.1 M Tris pH 8.5, and 1.5 M lithium sulfate in hanging drop format. Crystals were cryoprotected by the addition of 24% glycerol and flash-frozen in liquid nitrogen. Diffraction data were collected at the Advanced Photon Source NE-CAT beamline 24ID-C on 11 December 2021, with X-ray wavelength 0.97911 Å and at 100 K. Data were processed with the RAPD data-processing pipeline (https://github.com/RAPD/RAPD), which uses XDS[36] for data indexing and reduction, AIMLESS[37] for scaling, and TRUNCATE[38] for conversion to structure factors. We determined the structure by molecular replacement in PHASER[39], using a predicted structure from AlphaFold2 (ref. 40) as a search model. We manually rebuilt the initial model in COOT[41] and refined it in phenix.refine[42] using positional and individual *B* factor refinement (Table 3). The final model had good geometry, with 98.75% of residues in favored Ramachandran space, 1.25% allowed and 0% outliers. The overall MolProbity score was 0.92, and the MolProbity clash score was 1.7.

## Statistics and reproducibility
All light microscopy experiments were performed once, and each set of conditions that were directly compared (for example, phage-infected versus uninfected) were performed in the same experiment. For figure panels where single cells are shown (Figs. 1a,c–e, 2a,b and 4a and Extended Data Figs. 1b, 2a,b, 3 and 8a,b), these cells are representative of the cell population across at least three full fields of cells recorded in the microscope (in all cases, at least 20 cells were individually inspected). For quantitation of cellular characteristics (Fig. 4b–e and Extended Data Fig. 8c–e), at least 100 cells were examined for each sample. Sample sizes for quantitative characteristics were chosen to ensure at least 95% confidence in observing effect sizes over 10%.

GFP-Trap purifications and mass spectrometry analysis (Fig. 2c and Extended Data Fig. 5a) were performed once per sample. Protein coexpression assays were performed at least three times with consistent results (Fig. 2e,f and Extended Data Figs. 5b and 7d,e).

## Reporting summary
Further information on research design is available in the Nature Portfolio Reporting Summary linked to this article.

## Data availability
Mass spectrometry data are available at the PRIDE database (www.ebi.ac.uk/pride) under accession ID PXD041684. Final refined coordinates and reduced diffraction data for the structure of PA1C gp2 are available at the RCSB Protein Data Bank (www.rcsb.org) under accession ID 7UYX. Raw diffraction data for the structure of PA1C gp2 are available at the SBGrid Data Bank (data.sbgrid.org) under accession ID 908. Publicly accessible data used in this study include the genome sequence of phage PhiPA3 (NCBI RefSeq NC_028999.1) and annotated protein sequences for phages PhiPA3 (NCBI RefSeq YP_009217083.1 to YP_009217457.1), 201Phi2-1 gp2 (YP_001956728.1) and gp105 (YP_001956829.1), Psa21 gp2 (YP_010347551.1) and phage PA1C gp3 (QBX32150.1). Source data are provided with this paper.

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

## Acknowledgements
We thank members of the Pogliano and Corbett laboratories for helpful discussions and for critical reading of the manuscript. We acknowledge support from the National Institutes of Health (R01 GM129245 to J.P. and E.V.; R35 GM144121 to K.D.C.; and NIH shared instrumentation grant S10 OD021724) and the Howard Hughes Medical Institute Emerging Pathogens Initiative (to E.V., J.P. and K.D.C.). E.V. is a Howard Hughes Medical Institute Investigator. This work is based on research conducted at the Northeastern Collaborative Access Team beamlines, which are funded by the National Institute of General Medical Sciences from the National Institutes of Health (P30 GM124165). This research used resources of the Advanced Photon Source, a U.S. Department of Energy (DOE) Office of Science User Facility operated for the DOE Office of Science by Argonne National Laboratory under contract number DE-AC02-06CH11357.

## Author contributions
E.E. conceived the study and performed all cloning, microscopy, protein purification and preparation of samples for mass spectrometry, as well as data analysis and figure and manuscript preparation. A.D. performed X-ray crystallography data collection and analysis. Y.G. performed SEC–MALS analysis. K.T.N. and V.C. cloned and characterized GFP fusions of phage proteins. E.A. performed data analysis and provided input on manuscript preparation. M.G. performed mass spectrometry experiments and initial mass spectrometry data analysis. E.V. provided input on experimental design and interpretation. J.P. conceived the study, guided experimental design and data interpretation, provided funding, prepared figures and wrote the manuscript. K.D.C. guided experimental design and data interpretation, provided funding, prepared figures and wrote the manuscript.

## Competing interests
The authors declare no competing interests.

## Additional information
**Extended data** is available for this paper at https://doi.org/10.1038/s41594-023-01094-5.

**Correspondence and requests for materials** should be addressed to Joe Pogliano or Kevin D. Corbett.

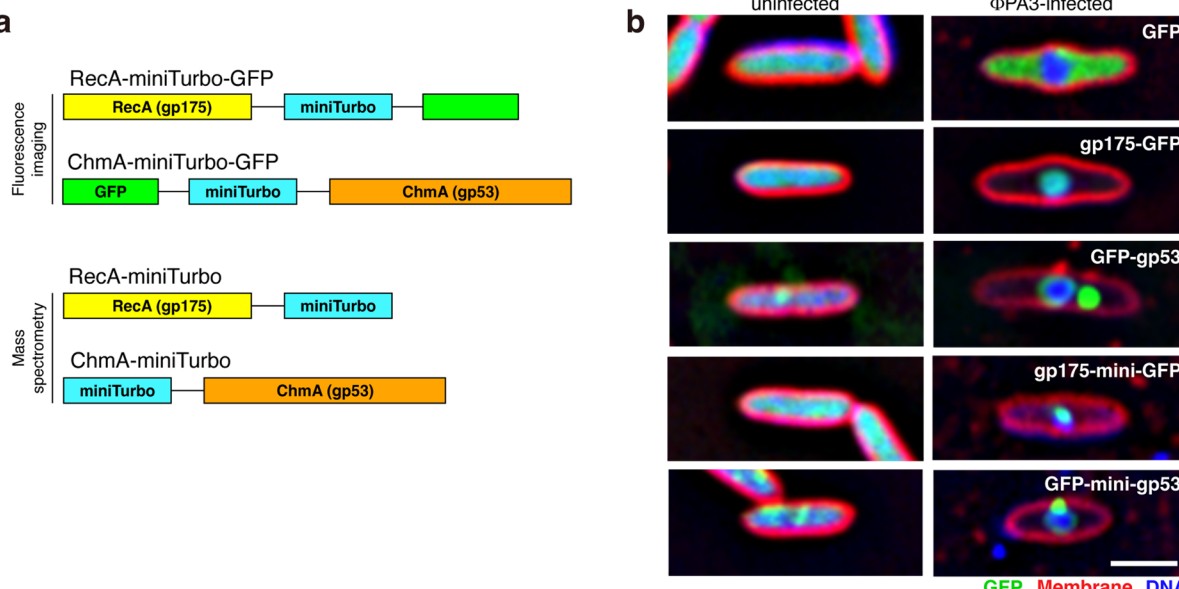

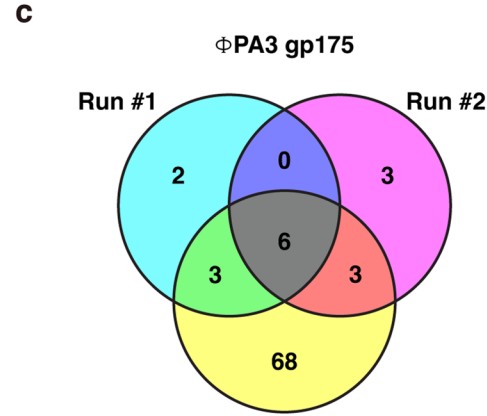

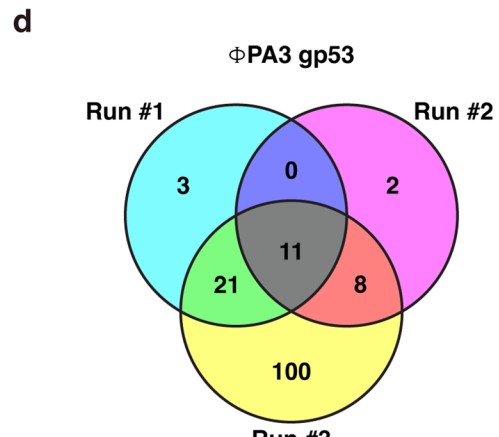

**Extended Data Fig. 1 | miniTurboID proximity labeling of phage nucleus-associated proteins. (a)** Construct design for localization and proximity labeling of ΦPA3 RecA (gp175) and ChmA (gp53) associated proteins. **(b)** Localization of GFP control and GFP- and GFP-miniTurboID tagged RecA (gp175) and ChmA (gp53) in ΦPA3-infected *P. aeruginosa* cells. GFP is shown in green,

FM4-64 (to visualize membranes) in red, and DAPI (to visualize nucleic acids) in blue. Scale bar = 2 μm. **(c, d)** Venn diagrams showing RecA (panel c) and ChmA (panel d) interacting proteins identified by miniTurboID labeling in three independent runs. See Tables 1–2 and Supplementary Tables 1-2.

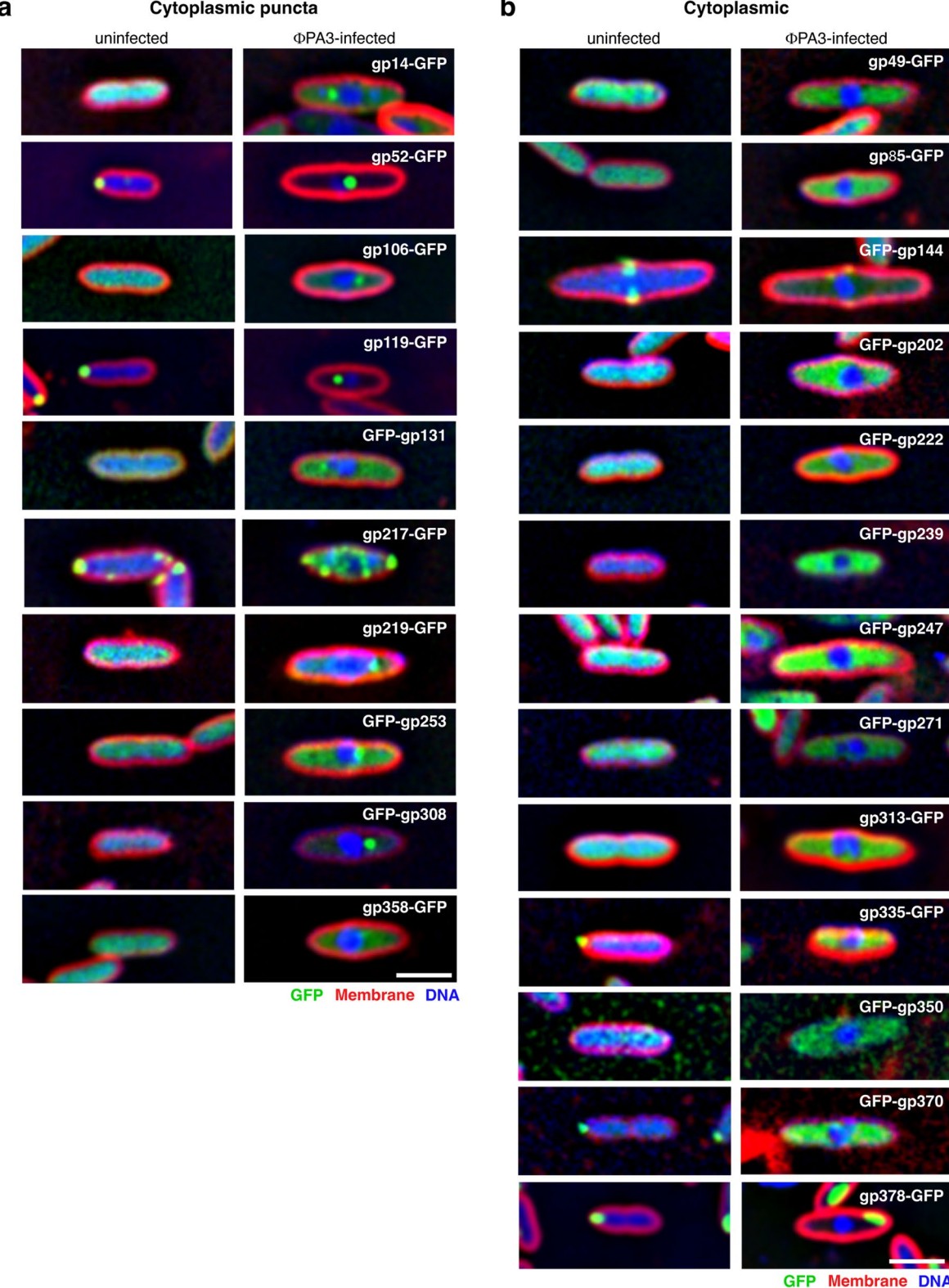

**Extended Data Fig. 2 | Localization analysis of RecA- and ChmA-interacting ΦPA3 proteins.** Subcellular localization of selected proteins identified by proximity labeling, with panel (**a**) showing proteins that localize as cytoplasmic puncta, and (**b**) showing proteins with diffuse cytoplasmic localization. See Table 3 for a collated list of localizations. GFP is shown in green, FM4-64 (to visualize membranes) in red, and DAPI (to visualize nucleic acids) in blue. Scale bar = 2 μm.

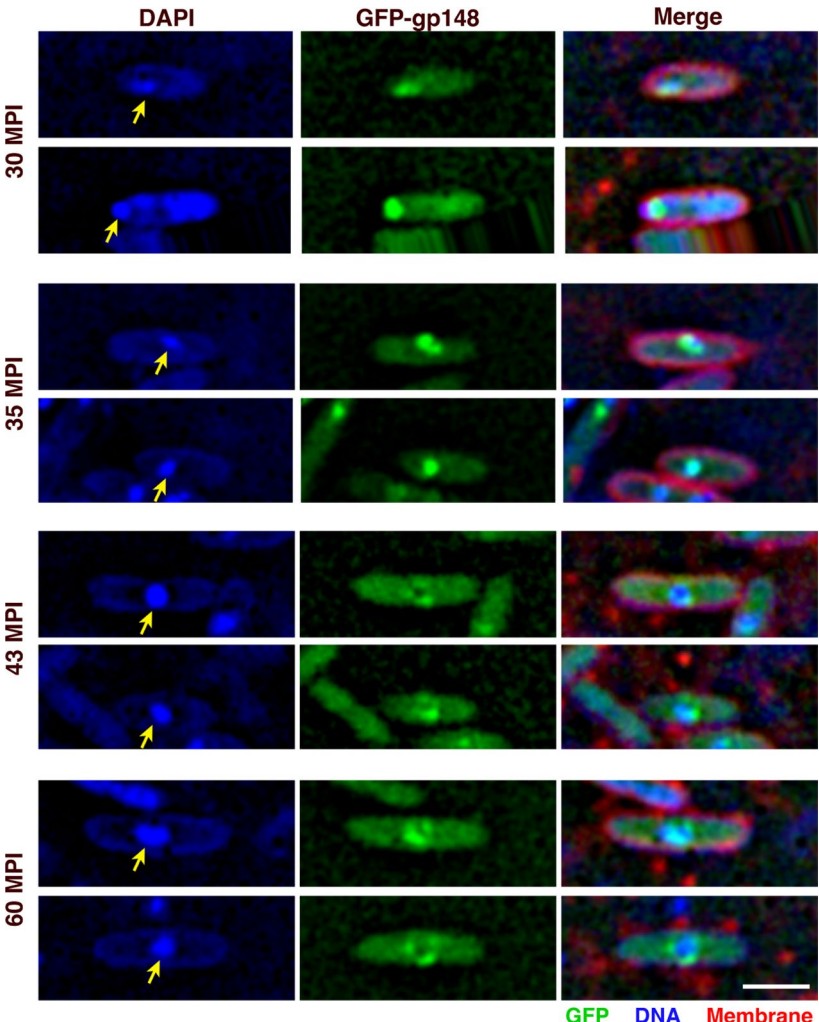

**Extended Data Fig. 3 | Early localization of gp148 to the phage nuclear shell.** Subcellular localization of GFP-gp148 in *P. aeruginosa* cells infected with phage ΦPA3, at the indicated times post infection (MPI: minutes post infection). Yellow arrows indicate the position of the phage nucleus. GFP is shown in green, FM4-64 (to visualize membranes) in red, and DAPI (to visualize nucleic acids) in blue. Scale bar = 2 μm.

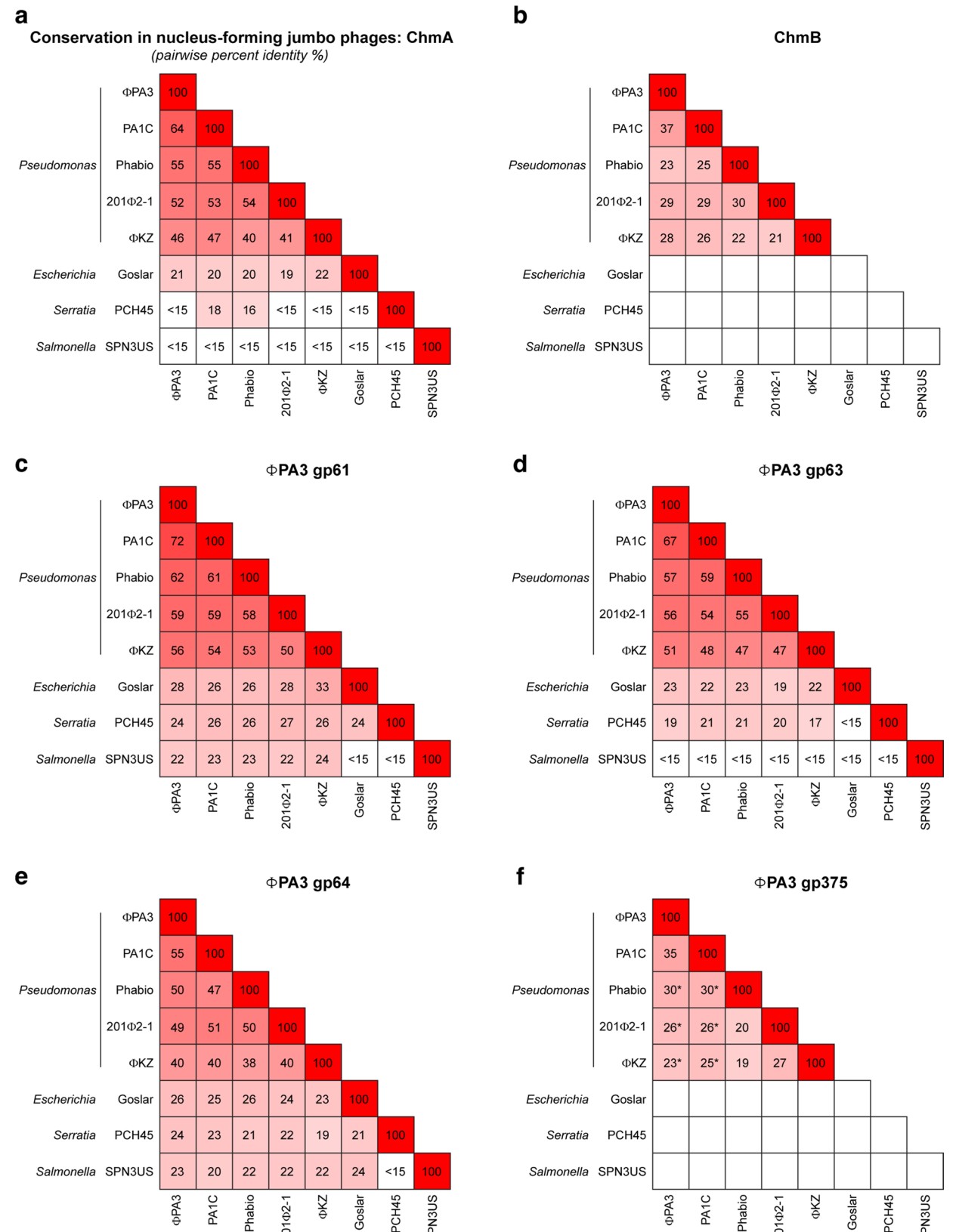

**Extended Data Fig. 4 | See next page for caption.**

**Extended Data Fig. 4 | Sequence analysis of nuclear-localized jumbo phage proteins.** (**a**) Sequence identity (expressed as percent identity) between selected jumbo phage ChmA proteins: ΦPA3 (NCBI Accession # YP_009217136.1), PA1C (QBX32206.1), Phabio (YP_010348051.1), 201Φ2-1 (YP_001956829.1), ΦKZ (NP_803620.1), Goslar (YP_009820873.1), PCH45 (QFP93061.1), SPN3US (YP_009153316.1). For all pairs of sequences, the 'Pairwise Alignment' tool in JalView was used to calculate % identity over the homologous region of each protein. For those identities reported as '<15%', this tool was unable to generate an alignment over a significant portion of the sequences. (**b**) Sequence identity between selected jumbo phage ChmB proteins: ΦPA3 (NCBI Accession # YP_009217084.1), PA1C (QBX32150.1), Phabio (YP_010347970.1), 201Φ2-1 (YP_001956728.1), ΦKZ (NP_803568.1). Homologs were not identified in Goslar, PCH45, or SPN3US. (**c**) Sequence identity between selected jumbo phage homologs of ΦPA3 gp61: ΦPA3 (NCBI Accession # YP_009217144.1), PA1C (QBX32215.1), Phabio (YP_010348072.1), 201Φ2-1 (YP_001956847.1), ΦKZ (NP_803633.1), Goslar (YP_009820861.1), PCH45 (QFP93121.1), SPN3US

(YP_009153323.1). (**d**) Sequence identity between selected jumbo phage homologs of ΦPA3 gp63: ΦPA3 (NCBI Accession # YP_009217146.1), PA1C (QBX32217.1), Phabio (YP_010348074.1), 201Φ2-1 (YP_001956849.1), ΦKZ (NP_803635.1), Goslar (YP_009820859.1), PCH45 (QFP93075.1), SPN3US (YP_009153325.1). (**e**) Sequence identity between selected jumbo phage homologs of ΦPA3 gp64: ΦPA3 (NCBI Accession # YP_009217147.1), PA1C (QBX32218.1), Phabio (YP_010348075.1), 201Φ2-1 (YP_001956850.1), ΦKZ (NP_803636.1), Goslar (YP_009820858.1), PCH45 (QFP93079.1), SPN3US (YP_009153326.1). (**f**) Sequence identity between selected jumbo phage homologs of ΦPA3 gp375: ΦPA3 (NCBI Accession # YP_009217454.1, 233 residues), PA1C (QBX32544.1, 250 residues), Phabio (YP_010348429.1, 736 residues), 201Φ2-1 (YP_001957174.1, 768 residues), ΦKZ (NP_803869.1, 646 residues). Homologs were not identified in Goslar, PCH45, or SPN3US. Asterisks indicate that these identities represent the homologous regions of dramatically different-length proteins.

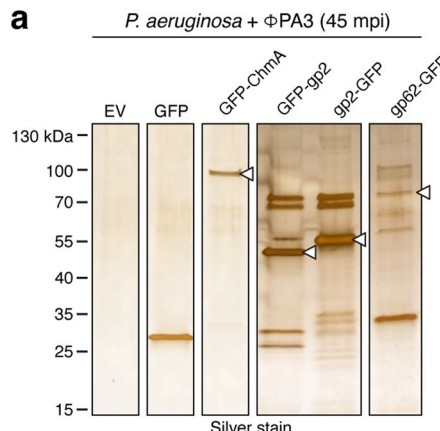

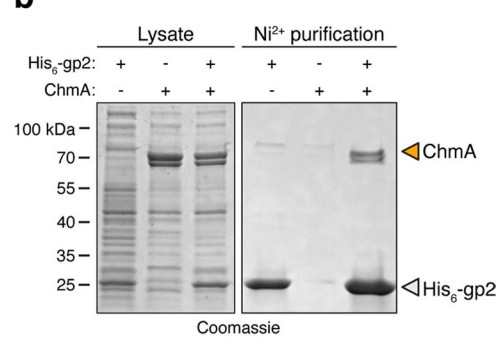

**Extended Data Fig. 5 | Protein-protein interaction analysis. (a)** Silver stained SDS-PAGE gels showing GFP pulldown results from GFP-tagged proteins expressed in ΦPA3-infected *P. aeruginosa* (45 minutes post infection). White arrowheads indicate the bait protein for each sample. (**b**) Coomassie blue-stained SDS-PAGE gel showing Ni²⁺ pulldown results from coexpression of 201Φ2-1 gp2 (His₆-tagged) and ChmA (gp105; untagged). ChmA appears as a doublet because of a second start codon at codon 33 of the gp105 gene.

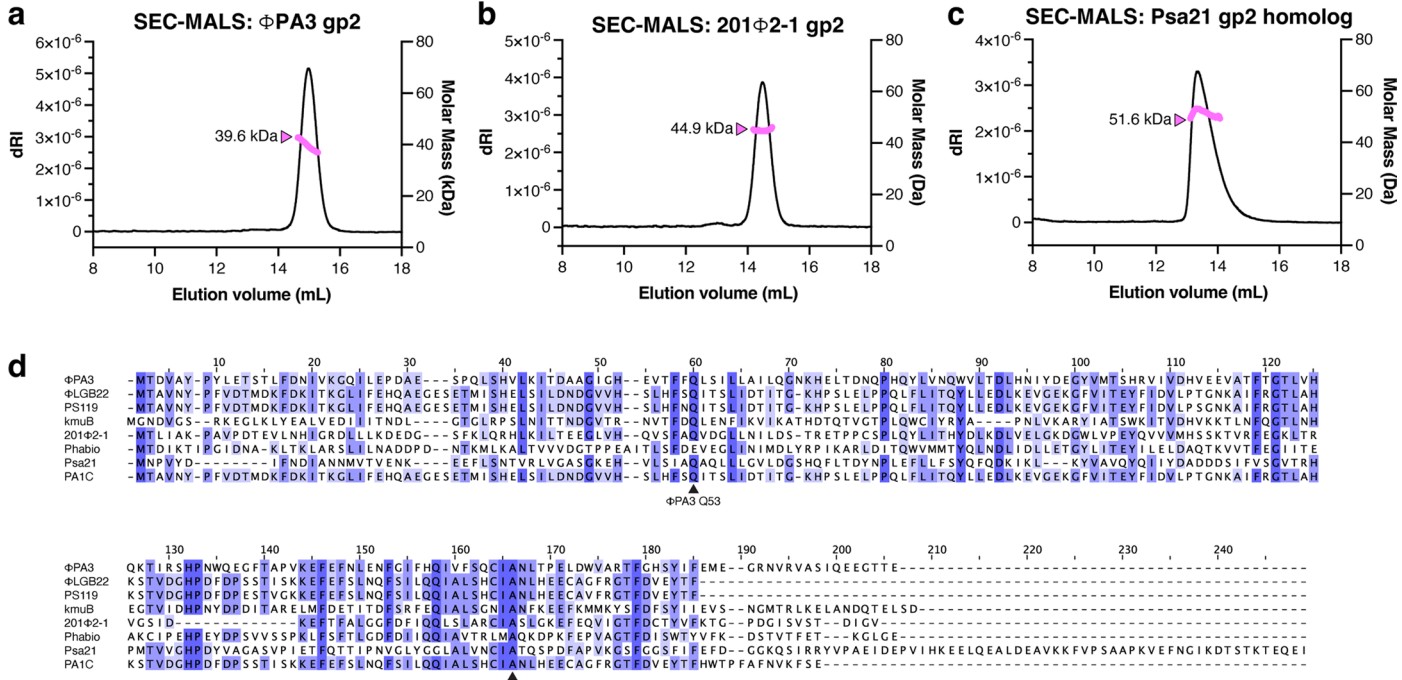

**Extended Data Fig. 6 | Biochemical and sequence analysis of jumbo phage gp2 proteins. (a)** Size exclusion chromatography coupled to multi-angle light scattering (SEC-MALS) analysis of ΦPA3 gp2. Measured molecular weight = 39.6 kDa; dimer molecular weight = 45 kDa. **(b)** SEC-MALS analysis of 201Φ201 gp2. Measured molecular weight = 44.9 kDa; dimer molecular weight = 39.8 kDa.

**(c)** SEC-MALS analysis of the gp2 homolog in phage Psa21 (gp3). Measured molecular weight = 51.6 kDa; dimer molecular weight = 50.6 kDa. **(d)** Sequence alignment of gp2 homologs in jumbo phage infecting *Pseudomonas*, showing the position of the highly conserved Q53 and A159 (ΦPA3 numbering) residues.

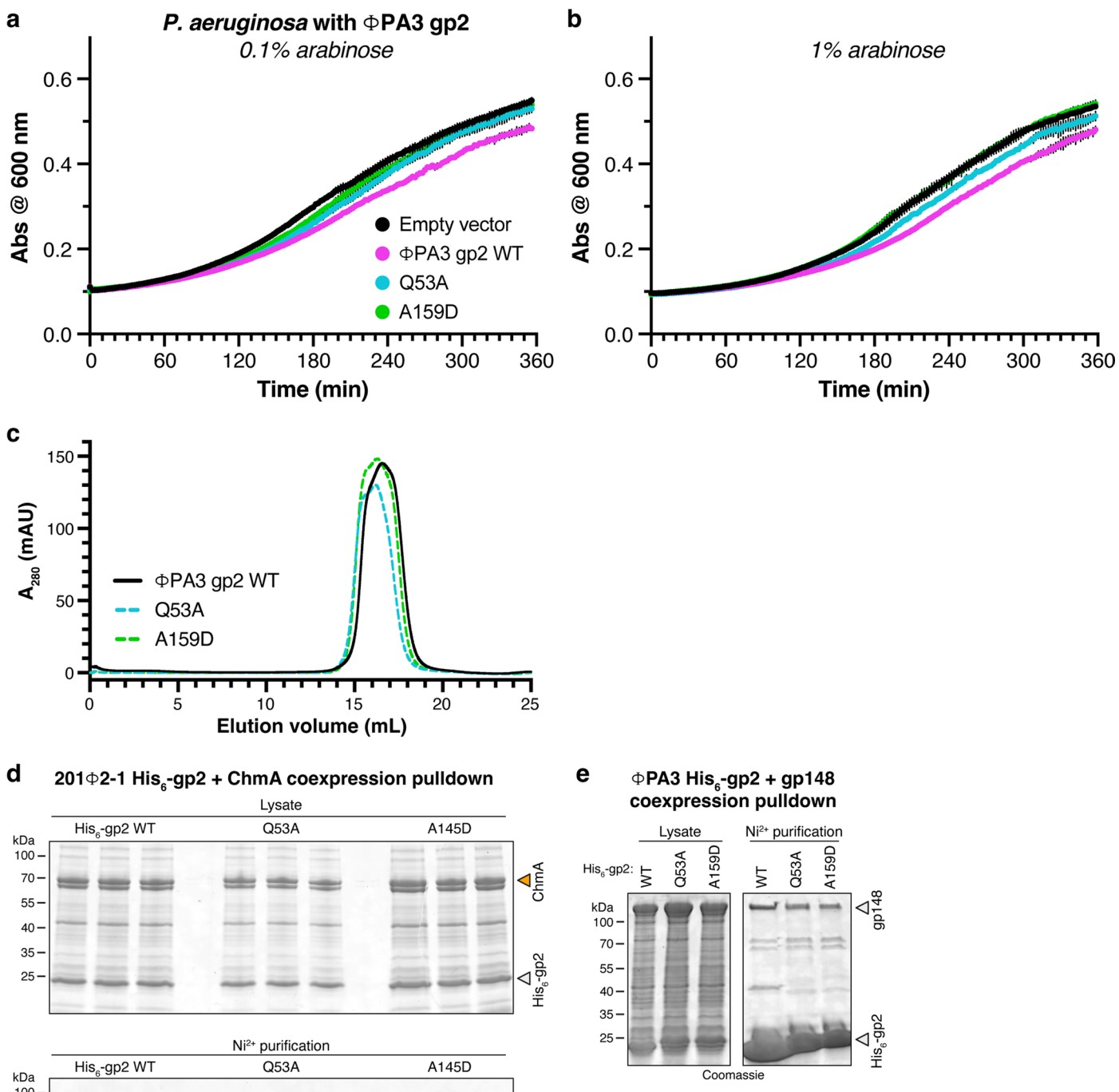

**Extended Data Fig. 7 | Effects of gp2 Q53A and A159D mutations. (a)** Growth curves of *P. aeruginosa* cells transformed with pHERD30T empty vector (black) or encoding ΦPA3 gp2 wild-type (violet), Q53A (cyan), or A159D (green). Growth media contained 0.1% arabinose for low-level expression. Datapoints represent the average of three independent replicates, and error bars (black) represent standard deviation. **(b)** Growth curves of *P. aeruginosa* cells transformed as in panel (a), with growth media containing 1% arabinose for high-level expression. Datapoints represent the average of three independent replicates, and error bars (black) represent standard deviation. **(c)** Size exclusion chromatography elution

profiles for ΦPA3 ΦPA3 gp2 wild-type (black solid line), Q53A (cyan dotted line), or A159D (green dotted line). **(d)** Ni²⁺ pulldown analysis of *E. coli*-coexpressed His₆-tagged 201Φ2-1 gp2 (wild-type (WT), Q53A, or A145D (equivalent to ΦPA3 gp2 A159D)) and full-length ChmA. Doublet bands for ChmA arise from a methionine codon at position 33 of the annotated gene. **(e)** Ni²⁺ pulldown analysis of *E. coli*-coexpressed His₆-tagged ΦPA3 gp2 (wild-type (WT), Q53A, or A159D) and portal (gp148). The ~40 kDa band in pulldown in the Ni²⁺ purification lane for WT gp2 is an unknown contaminant.

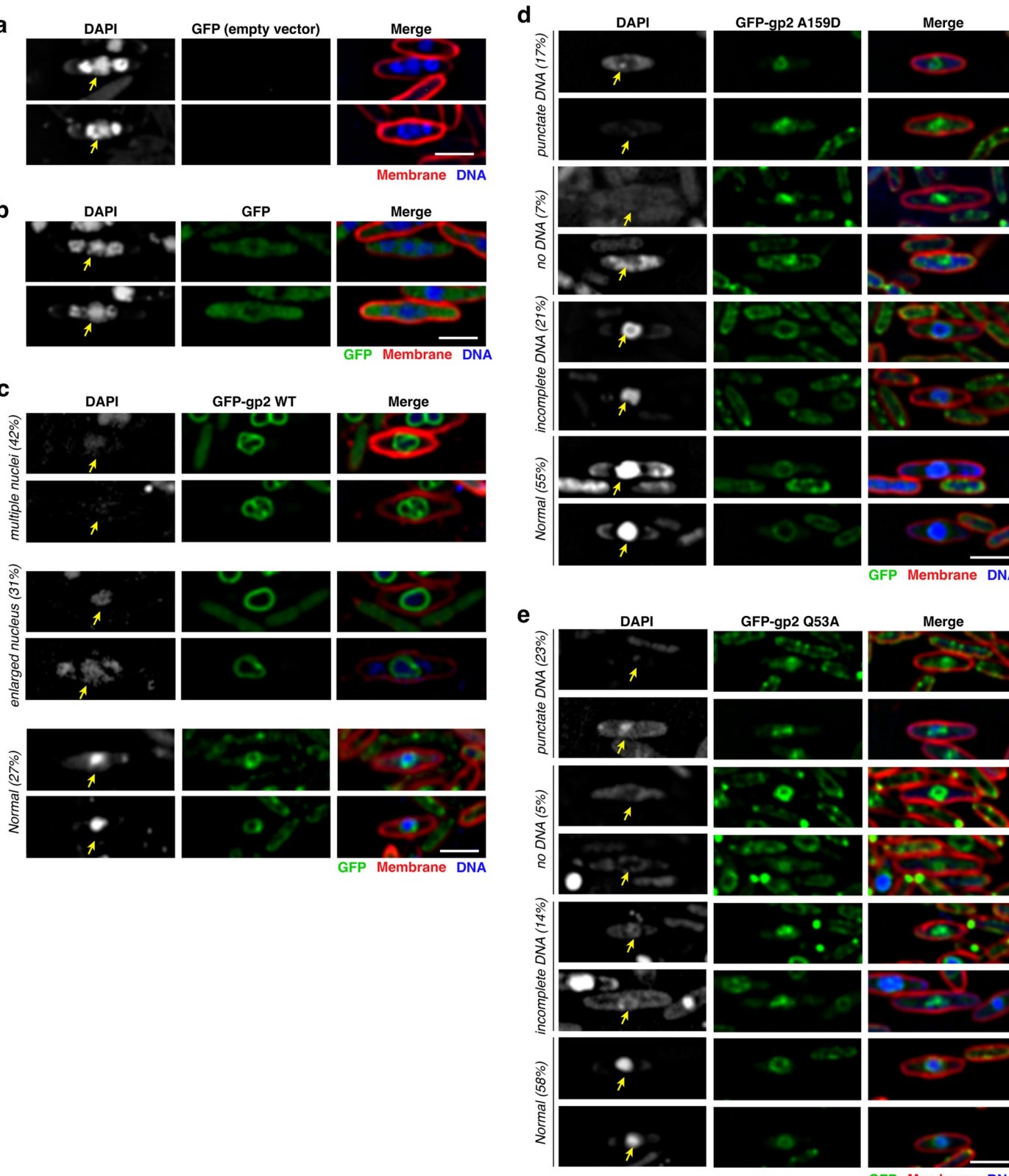

**Extended Data Fig. 8 | Overexpression of mutant gp2 proteins affect nuclear shell formation and morphology.** (**a**) Microscopy of ΦPA3-infected *P. aeruginosa* cells transformed with empty vector. Yellow arrow indicates the position of the phage nucleus. The two DAPI staining bodies bracketing the phage nucleus are the phage bouquets. GFP is shown in green, FM4-64 (to visualize membranes) in red, and DAPI (to visualize nucleic acids) in blue. Scale bar = 2 μm. (**b**) Microscopy of ΦPA3-infected *P. aeruginosa* cells expressing GFP. Scale bar = 2 μm. (**c**) Microscopy of ΦPA3-infected *P. aeruginosa* cells expressing

GFP-tagged wild-type gp2. Common phenotypes and their percentages in the analyzed sample (n = 100 cells) are shown. Scale bar = 2 μm. (**d**) Microscopy of ΦPA3-infected *P. aeruginosa* cells expressing GFP-tagged gp2 A159D. Common phenotypes and their percentages in the analyzed sample (n = 100 cells) are shown. Scale bar = 2 μm. (**e**) Microscopy of ΦPA3-infected *P. aeruginosa* cells expressing GFP-tagged gp2 Q53A. Common phenotypes and their percentages in the analyzed sample (n = 100 cells) are shown. Scale bar = 2 μm.

Pogliano, Joe

# Reporting Summary

## Statistics

For all statistical analyses, confirm that the following items are present in the figure legend, table legend, main text, or Methods section.

| n/a | Confirmed | |
|---|---|---|
| ☐ | ☒ | The exact sample size (*n*) for each experimental group/condition, given as a discrete number and unit of measurement |
| ☐ | ☒ | A statement on whether measurements were taken from distinct samples or whether the same sample was measured repeatedly |
| ☐ | ☒ | The statistical test(s) used AND whether they are one- or two-sided *Only common tests should be described solely by name; describe more complex techniques in the Methods section.* |
| ☒ | ☐ | A description of all covariates tested |
| ☐ | ☒ | A description of any assumptions or corrections, such as tests of normality and adjustment for multiple comparisons |
| ☐ | ☒ | A full description of the statistical parameters including central tendency (e.g. means) or other basic estimates (e.g. regression coefficient) AND variation (e.g. standard deviation) or associated estimates of uncertainty (e.g. confidence intervals) |
| ☐ | ☒ | For null hypothesis testing, the test statistic (e.g. *F*, *t*, *r*) with confidence intervals, effect sizes, degrees of freedom and *P* value noted *Give P values as exact values whenever suitable.* |
| ☒ | ☐ | For Bayesian analysis, information on the choice of priors and Markov chain Monte Carlo settings |
| ☒ | ☐ | For hierarchical and complex designs, identification of the appropriate level for tests and full reporting of outcomes |
| ☐ | ☒ | Estimates of effect sizes (e.g. Cohen's *d*, Pearson's *r*), indicating how they were calculated |

*Our web collection on statistics for biologists contains articles on many of the points above.*

## Software and code

Policy information about availability of computer code

| Data collection | DeltaVision softWoRx v. 7 |
|---|---|
| Data analysis | DeltaVision softWoRx v. 7, RAPD data-processing pipeline (https://github.com/RAPD/RAPD), COOT v. 0.9, Phenix package v. 1,2, PyMOL v. 2.5 , ASTRA v. 8, Peaks Studio v. 8.5, ImageJ v. 2, and GraphPad Prism v. 9 were used for this study |

For manuscripts utilizing custom algorithms or software that are central to the research but not yet described in published literature, software must be made available to editors and reviewers. We strongly encourage code deposition in a community repository (e.g. GitHub). See the Nature Portfolio guidelines for submitting code & software for further information.

## Data

Policy information about availability of data

All manuscripts must include a data availability statement. This statement should provide the following information, where applicable:
- Accession codes, unique identifiers, or web links for publicly available datasets
- A description of any restrictions on data availability
- For clinical datasets or third party data, please ensure that the statement adheres to our policy

Mass spectrometry data is available at the PRIDE database under accession number PXD041684. Final refined coordinates and reduced diffraction data for the structure of PA1C gp2 is available at the RCSB Protein Data Bank (www.rcsb.org) under accession ID 7UYX. Raw diffraction data for the structure of PA1C gp2 is available at the SBGrid Data Bank (data.sbgrid.org) under accession ID 908. Publicly accessible data used in this study includes the genome sequence of phage

PhiPA3 (NCBI RefSeq NC_028999.1), Annotated protein sequences of phage PhiPA3 (NCBI RefSeq YP_009217083.1 through YP_009217457.1), Annotated protein sequences of phage 201Phi2-1 gp2 (YP_001956728.1) and gp105 (YP_001956829.1), annotated protein sequence of phage Psa21 gp2 (YP_010347551.1), and annotated protein sequence of phage PA1C gp3 (QBX32150.1).

# Human research participants

Policy information about studies involving human research participants and Sex and Gender in Research.

| Reporting on sex and gender | N/A |
| --- | --- |
| Population characteristics | N/A |
| Recruitment | N/A |
| Ethics oversight | N/A |

Note that full information on the approval of the study protocol must also be provided in the manuscript.

# Field-specific reporting

Please select the one below that is the best fit for your research. If you are not sure, read the appropriate sections before making your selection.

☒ Life sciences       ☐ Behavioural & social sciences       ☐ Ecological, evolutionary & environmental sciences

For a reference copy of the document with all sections, see nature.com/documents/nr-reporting-summary-flat.pdf

# Life sciences study design

All studies must disclose on these points even when the disclosure is negative.

| Sample size | For descriptive characterization of cellular phenotypes, at least 20 cells were visually examined for each sample. For quantitation of cellular characteristics, at least 100 cells were examined for each sample. Sample sizes for quantitative characteristics were chosen to ensure at least 95% confidence in observing effect sizes over 10%. |
| --- | --- |
| Data exclusions | No data were excluded from the analysis. |
| Replication | Subcellular localization and other phenotypic analyses were reproducible across a large number of cells (n=20-100 depending on experiment). Protein coexpression was performed three times on different days, and the noted interactions were reproducible. |
| Randomization | Samples were not randomized; no experiments involved covariates. |
| Blinding | For phenotype quantification, blinding was not required because quantitative criteria were established for each phenotype and applied in an unbiased manner across all cells in a given field. |

# Reporting for specific materials, systems and methods

We require information from authors about some types of materials, experimental systems and methods used in many studies. Here, indicate whether each material, system or method listed is relevant to your study. If you are not sure if a list item applies to your research, read the appropriate section before selecting a response.

## Materials & experimental systems

| n/a | Involved in the study |
| --- | --- |
| ☒ ☐ | Antibodies |
| ☒ ☐ | Eukaryotic cell lines |
| ☒ ☐ | Palaeontology and archaeology |
| ☒ ☐ | Animals and other organisms |
| ☒ ☐ | Clinical data |
| ☒ ☐ | Dual use research of concern |

## Methods

| n/a | Involved in the study |
| --- | --- |
| ☒ ☐ | ChIP-seq |
| ☒ ☐ | Flow cytometry |
| ☒ ☐ | MRI-based neuroimaging |

