## [Peer Review File · Nature Structural & Molecular Biology]

Peer Review Information

Manuscript Title: Identification of the bacteriophage nucleus protein interaction network

Corresponding author name(s): Joe Pogliano, Kevin D. Corbett

Reviewer Comments & Decisions:

Decision Letter, initial version:
--

Message: 22nd Sep 2022

Dear Dr. Corbett,

Thank you again for submitting your manuscript "Identification of the bacteriophage nucleus protein interaction network". I sincerely apologize for the delay in responding, which resulted from the difficulty in obtaining suitable referee reports. Nevertheless, we now have comments (below) from the 3 reviewers who evaluated your paper. In light of those reports, we remain interested in your study and would like to see your response to the comments of the referees, in the form of a revised manuscript.

You will see that while all reviewers appreciate the work, that reviewers #2 and #3 raise some mechanistic questions about the role of the mutant ChmB. Please be sure to address/respond to all concerns of the referees in full in a point-by-point response and highlight all changes in the revised manuscript text file. If you have comments that are intended for editors only, please include those in a separate cover letter.

We expect to see your revised manuscript within 6 weeks. If you cannot send it within this time, please contact us to discuss an extension; we would still consider your revision, provided that no similar work has been accepted for publication at NSMB or published elsewhere.

Reporting Summary:

Please note that all key data shown in the main figures as cropped gels or blots should be presented in uncropped form, with molecular weight markers. These data can be aggregated into a single supplementary figure item. While these data can be displayed in a relatively informal style, they must refer back to the relevant figures. These data should be submitted with the final revision, as source data, prior to acceptance, but you may want to start putting it together at this point.

Data availability: this journal strongly supports public availability of data. All data used in accepted papers should be available via a public data repository, or alternatively, as Supplementary Information. If data can only be shared on request, please explain why in your Data Availability Statement, and also in the correspondence with your editor. Please note that for some data types, deposition in a public repository is mandatory - more

information on our data deposition policies and available repositories can be found below:
<https://www.nature.com/nature-research/editorial-policies/reporting-standards#availability-of-data>

[REDACTED]

Sincerely,
Sara

Sara Osman, Ph.D.
Associate Editor
Nature Structural & Molecular Biology

Reviewers' Comments:

Reviewer #1:

Remarks to the Author:

The study by Enustun et al. described in this manuscript represents a significant breakthrough in understanding the structure and molecular mechanics of "phage nuclei" formed by some jumbo phages of *P. aeruginosa*. A new phage protein, gp2, apparently playing a central role in "nucleus" organisation was described here and studied in much detail. The study uses state of the art methodology, with all main conclusions about proteins interacting with/forming nucleus structure being supported by more than one experimental approach (e.g. proximity labelling, sub-cellular co-localisation, co-precipitation, site-directed mutagenesis, etc.). All results are very clearly described and beautifully illustrated with a set of main and supplementary figures. I recommend this manuscript for publication after some minor improvement (see below).

Specific comments:

Line 80. What expression vector was used to transform *P. aeruginosa*? Not clear from the methods section. Table S6 (plasmid constructs coding for fusion proteins) is missing.

Line 121. A supplementary figure with protein sequence alignments would be handy to illustrate sequence conservation.

Line 151. "Some of these proteins were also identified in pulldowns with other GFP-tagged proteins, 152 suggesting that the proteins may simply be highly abundant in cell lysates" - Could the same be also true for MCP (gp136)?

Line 186. Please explain why a homologue from phage PA1C and not Φ PA3 gp2 itself was used here.

Line 245. NTS and CTS need to be explained at first use.

Line 290 and throughout the Methods section: replace Φ Phi2-1 with Φ 2-1 (otherwise it's "Phi" twice).

Spelling/grammar check is required.

Reviewer #2:

Remarks to the Author:

Upon infecting a host cell, some jumbo phages form a proteinaceous envelope around their DNA (pseudonucleus) which protects it from host defense systems. Various phage proteins are imported into the pseudonucleus and phage mRNAs are exported to the cytosol to be translated. How this transport occurs mechanistically has been elusive. In the present study, Enustun and colleagues provide insight into this intriguing biological phenomenon. By using proximity labeling and localization mapping, they identify six phage proteins associated with the nuclear shell. They further demonstrate that one of them (named ChmB) directly interacts with the ChmA, the main component of the nuclear envelope. Moreover, they solve the crystal structure of ChmB and find that this protein is required for the proper pseudonucleus formation.

The paper is comprehensive and clearly written. The results are compelling and represent an important step forward in understanding the life cycle of pseudonucleus-forming phages.

Major comments:

1. Terminology: In general, the term transport (line 30) implies an active process, usually energy-dependent (for example, consuming ATP). As far as we can see, none of the

identified interaction partners are predicted ATPases, hence it is very likely that this process is an energy-free translocation (or channeling), so we recommended it be referred to as translocation. Similarly, while we understand the authors' excitement with phages, these are not "organisms" (line 34), so please correct this.

2. The authors show that ChmB Q53A and A159A mutants are inactive (i.e. their overproduction leads to aberrant phage nucleus formation) but offer no insight as to why. The authors should perform pull down experiments with GFP-tagged mutated proteins and compare the results to those obtained for wt ChmB. Do these mutant proteins lose the ability to interact with ChmA or another interaction partner? Are they still able to dimerize, i.e. do they pull down wt ChmB? Looking at the position of these conserved residues in the structure, can anything be inferred about their contribution to dimerization?

3. Fig. S1C and D shows that the 3rd replicate of the proximity labeling experiment yielded substantially a higher number of putative interaction partners compared to the first two replicates. What is the explanation for this and what does this tell us about robustness / reproducibility / false positive discovery rate of the method? This needs to be discussed.

Minor comments:

4. The authors speculate that ChmB may contribute to protein trafficking into the pseudonucleus. To further support this claim, the authors may consider to look at pseudonucleus localization defects upon expression of mutated ChmB, but this is really optional.

5. Fig. 1B is misleading as it suggests that the miniTurboID is fused to the C terminus of both RecA and ChmA, which is not the case, according to Fig. S1A. To avoid confusion, the scheme in Fig. 1B should show miniTurboID fused to the N terminus of ChmA and labeled as miniTurbo-ChmA.

6. Figure S1: The authors should state in the main text that for the MS experiments no GFP was fused to the constructs and explain why. They should also clarify that the viability of the miniTurbo constructs was validated with the GFP fusion but then the truncated variant without GFP was used for MS.

7. Lines 164-169: Does ChmA from phi-PA3 also self-assemble into closed structures? And how conserved is ChmA between phi-PA3 and 201-phi-2-1? These two points require more elaboration.

8. Line 193 onwards: The authors should explain why they are certain that the overexpression of wt or mutated variants of ChmB do not cause a growth defect that would affect subsequent observations with the pseudonucleus.

Reviewer #3:

Remarks to the Author:

In this manuscript, the authors identify several proteins of jumbo phage fPA3 that interact with the ChmA protein, the major component of the pseudonucleus that compartmentalises phage DNA inside the infected cells and is the site of phage DNA replication and transcription. They show that at least one of the identified interactors of

ChmA, a conserved jumbo protein they term ChmB, also binds to phage portal protein, suggesting that it may coordinate phage virion morphogenesis and DNA loading at the outer surface of the pseudonucleus. They present the high resolution structure of ChmB, a protein with a unique fold, and show that overproduction of ChmB with mutations in the conserved surface residues prevents pseudonucleus formation in phage-infected cells. Overproduction of wild-type ChmB leads to enlarged pseudonuclei.

These are interesting observation and experiments are of high quality. Experiments of ChmB overproduction are potentially the most interesting for the understanding of pseudonucleus assembly and its role in phage progeny production. However, these experiments appear to be preliminary, since the functional consequences of overproduction of wild-type or mutant ChmB are not studied further. In particular, why are pseudonuclei increased in size upon wild-type protein overproduction? Why do nuclei fail to form upon mutant proteins overproduction? Are these direct or indirect effects? How is expression of phage early genes affected when cells overproducing ChmB are infected? How does the protein interaction network change upon ChmB overproduction? I believe these questions need to be addressed prior to publication in NSMB.

Author Rebuttal to Initial comments

We thank the reviewers for their insightful comments, questions, and suggestions. Below we address each reviewer comment in **blue text**. In addition, the attached manuscript has all substantive changes marked in **red text**.

Reviewer #1:

The study by Enustun et al. described in this manuscript represents a significant breakthrough in understanding the structure and molecular mechanics of "phage nuclei" formed by some jumbo phages of *P. aeruginosa*. A new phage protein, gp2, apparently playing a central role in "nucleus" organisation was described here and studied in much detail. The study uses state of the art methodology, with all main conclusions about proteins interacting with/forming nucleus structure being supported by more than one experimental approach (e.g. proximity labelling, sub-cellular co-localisation, co-precipitation, site-directed mutagenesis, etc.). All results are very clearly described and beautifully illustrated with a set of main and supplementary figures. I recommend this manuscript for publication after some minor improvement (see below).

Specific comments:

Line 80. What expression vector was used to transform *P. aeruginosa*? Not clear from the methods section. Table S6 (plasmid constructs coding for fusion proteins) is missing.

All protein expression in *P. aeruginosa* and *P. chlororaphis* used the vector pHERD30T. We now note this in the **Methods** section, and in **Extended Data Table S6** (now included; we apologize for this oversight).

Line 121. A supplementary figure with protein sequence alignments would be handy to illustrate sequence conservation.

We agree with the reviewer. Instead of simply presenting sequence alignments, however, we sought to present this sequence conservation data in a more visually informative way. The new **Extended Data Figure 4** presents pairwise sequence identity data for six proteins (ChmA, ChmB/gp2, gp61, gp63, gp64, and gp375) across eight nucleus-forming jumbo phages: five that infect *Pseudomonas*, and one each that infect *Escherichia*, *Serratia*, and *Salmonella*. These color-coded plots show how much more divergent gp2 homologs are compared to other phage nucleus-associated proteins, likely explaining why we are unable to detect gp2 homologs in nucleus-forming jumbo phages outside the group that infect *Pseudomonas*.

Line 151. "Some of these proteins were also identified in pulldowns with other GFP-tagged proteins, suggesting that the proteins may simply be highly abundant in cell lysates" - Could the same be also true for MCP (gp136)?

This sentence was regrettably vague in the original manuscript. In the revised manuscript, we have rephrased this sentence to clarify that two of the 14 predicted phage structural

proteins identified in gp2 pulldowns (gp213 and gp11) are also found in pulldowns of the non-virion RNA polymerase subunit gp62, suggesting that these proteins may have been identified because of high abundance in infected cells.

Line 186. Please explain why a homologue from phage PA1C and not Φ PA3 gp2 itself was used here.

We purified and set initial crystal trays for several gp2 homologs, including Φ PA3 gp2. While Φ PA3 gp2 did form crystals, these crystals did not diffract X-rays to the resolution needed for structure determination. In contrast, gp2 from PA1C crystallized and diffracted X-rays to high resolution. It is not uncommon for closely related homologs to behave differently in crystallization trials, and therefore it is a common strategy to screen several homologs before finding one that crystallizes readily and diffracts to high resolution. Based on sequence identity and AlphaFold structure predictions, we expect that the structure of Φ PA3 gp2 is very similar to that of PA1C gp2.

Line 245. NTS and CTS need to be explained at first use.

These terms are defined in the **Results** section titled “gp2 interacts directly with ChmA and the phage portal protein”, but we have added further clarification to the **Discussion** as well to ensure clarity.

Line 290 and throughout the Methods section: replace Φ Phi2-1 with Φ 2-1 (otherwise it's "Phi" twice)

We have changed these terms, which unfortunately arose from a find-and-replace error.

Spelling/grammar check is required.

We have carefully checked all spelling and grammar in the revised manuscript.

Reviewer #2:

Upon infecting a host cell, some jumbo phages form a proteinaceous envelope around their DNA (pseudonucleus) which protects it from host defense systems. Various phage proteins are imported into the pseudonucleus and phage mRNAs are exported to the cytosol to be translated. How this transport occurs mechanistically has been elusive. In the present study, Enustun and colleagues provide insight into this intriguing biological phenomenon. By using proximity labeling and localization mapping, they identify six phage proteins associated with the nuclear shell. They further demonstrate that one of them (named ChmB) directly interacts with the ChmA, the main component of the nuclear envelope. Moreover, they solve the crystal structure of ChmB and find that this protein is required for the proper pseudonucleus formation.

The paper is comprehensive and clearly written. The results are compelling and represent an important step forward in understanding the life cycle of pseudonucleus-forming phages.

Major comments:

1. Terminology: In general, the term transport (line 30) implies an active process, usually energy-dependent (for example, consuming ATP). As far as we can see, none of the identified interaction partners are predicted ATPases, hence it is very likely that this process is an energy-free translocation (or channeling), so we recommended it be referred to as translocation. Similarly, while we understand the authors' excitement with phages, these are not "organisms" (line 34), so please correct this.

We have changed the term "transport" to "translocation" throughout the manuscript, as the reviewer is correct that we have not demonstrated active transport of either mRNA or protein through the nuclear shell. As a note, our observation that most nuclear-localized proteins are clearly preferentially localized to the nucleus rather than distributed throughout the cell suggests that protein import is likely an active, energy-consuming process. Similarly, passive diffusion of long mRNAs through even a sizable pore is difficult to imagine, and we anticipate mRNA export is also an active process (perhaps linked directly to transcription). However, these points are better left for future work that directly demonstrates them.

Similarly, we have removed the term "organisms" from the first sentence of the introduction and replaced it with "entities".

2. The authors show that ChmB Q53A and A159A mutants are inactive (i.e. their overproduction leads to aberrant phage nucleus formation) but offer no insight as to why. The authors should perform pull down experiments with GFP-tagged mutated proteins and compare the results to those obtained for wt ChmB. Do these mutant proteins lose the ability to interact with ChmA or another interaction partner? Are they still able to dimerize, i.e. do they pull down wt ChmB? Looking at the position of these conserved residues in the structure, can anything be inferred about their contribution to dimerization?

We chose Q53A and A159D mutants specifically because these residues are highly conserved yet surface-exposed and not involved in dimer interactions, and were therefore unlikely to affect the structure of gp2. To validate this hypothesis, we have added further data showing that both mutants are readily purified and do not affect the oligomeric state of gp2 in vitro (**Extended Data Figure 7c**).

The reviewer is perfectly correct to point out the need for pulldowns with gp2 and ChmA/portal to determine the effects of the mutations on these interactions. We now include these experiments, and find that both mutations mildly compromise binding to ChmA (**Fig. 4f**), but do not affect binding to the portal protein (**Fig. 4g**). With respect to ChmA binding, we note that neither mutation completely compromised ChmA binding in vitro, nor did the

mutants compromise nuclear shell binding in infected cells (**Fig. 4a**). Nonetheless, this effect is noteworthy and is now discussed in the revised manuscript.

We did previously consider a mass spectrometry experiment like the reviewer suggests. We decided not to pursue this experiment for two reasons. First, because there is phage-encoded wild-type gp2 expressed in the infected cells, mutant gp2 could dimerize with wild-type gp2 and pull down its native binding partners. Second, given the high variability in mass spectrometry experiments, we would not be confident that loss of an observed interaction meant that the interaction was truly lost through the mutation, or simply that the binding partner wasn't detected by mass spectrometry. We therefore decided not to pursue this experiment.

Based on the gp2 mutants' phenotype in infected cells and the finding that they retain their ability to dimerize, interact with ChmA (partially) and the portal protein in vitro, and localize to the phage nuclear shell, we propose that the mutants disrupt binding to an as-yet undefined interaction partner involved in an early step in the phage life cycle. We have updated and clarified the text to reflect this idea.

3. Fig. S1C and D shows that the 3rd replicate of the proximity labeling experiment yielded substantially a higher number of putative interaction partners compared to the first two replicates. What is the explanation for this and what does this tell us about robustness / reproducibility / false positive discovery rate of the method? This needs to be discussed.

The third set of replicates of our proximity labeling experiments was done at a different time than the first two, which were performed on biological replicates on the same day. For all samples (including controls), the third replicate showed more identified peptides and proteins labeled, likely due to higher overall specific and nonspecific activity of the biotin ligase resulting from experimental variations (time, temperature). Because hits were identified as those peptides showing at least 8-fold higher signal in experimental samples than in the *average* of the three GFP controls (\log_2 value of 3 or higher), more proteins achieved this level in the third replicate of each experiment than the other two (high experiment-to-experiment variation precluded comparisons between individual samples and matched controls). Most of the additional proteins identified in the third replicates are likely nonspecific hits due to the promiscuity of the biotin ligase, combined with the relatively high level of the fusion proteins and the small volume of the bacterial cell.

The reviewer is correct that overall, the method is not highly quantitative or reproducible on these samples. This is why we applied the $\log_2 \geq 3$ test, then ranked hits by overall peak area (see **Tables 1** and **2**), then tested the entire resulting set of top hits for their subcellular localization in phage-infected cells prior to claiming anything about these proteins' interactions or localization. As **Table 3** shows, a significant fraction of the 42 proteins identified by mass spectrometry and then tested in this assay showed cytoplasmic localization. In later experiments, we focused only on those hits for which we verified nuclear shell localization.

We do not claim to have identified the complete set of phage shell proteins or nuclear-localized proteins. Instead we see our current mass spectrometry experiments as an important lead-generation experiment that, because of its overall high noise and relative incompleteness, needs to be comprehensively followed up by functional experiments.

Minor comments:

4. The authors speculate that ChmB may contribute to protein trafficking into the pseudonucleus. To further support this claim, the authors may consider to look at pseudonucleus localization defects upon expression of mutated ChmB, but this is really optional.

We thank the reviewer for the suggestion. While we considered this experiment, the existence of phage-encoded wild-type ChmB alongside the mutant proteins in infected cells would complicate the analysis of this experiment.

5. Fig. 1B is misleading as it suggests that the miniTurboID is fused to the C terminus of both RecA and ChmA, which is not the case, according to Fig. S1A. To avoid confusion, the scheme in Fig. 1B should show miniTurboID fused to the N terminus of ChmA and labeled as miniTurbo-ChmA.

We apologize for the misleading figure. We have altered **Fig. 1b** to more accurately and clearly depict each fusion that was used.

6. Figure S1: The authors should state in the main text that for the MS experiments no GFP was fused to the constructs and explain why. They should also clarify that the viability of the miniTurbo constructs was validated with the GFP fusion but then the truncated variant without GFP was used for MS.

We have added further clarification of these points in the **Results** section.

7. Lines 164-169: Does ChmA from phi-PA3 also self-assemble into closed structures? And how conserved is ChmA between phi-PA3 and 201-phi-2-1? These two points require more elaboration.

We have rephrased this sentence to reflect the growing consensus that all nucleus-forming jumbo phages form a nuclear shell primarily composed of the ChmA protein, which is conserved among these phages and forms a conserved square lattice in diverse jumbo phages. Our prior work has demonstrated this finding for the *Pseudomonas* phage 201Φ2-1 and the *E. coli* phage Goslar, whose ChmA proteins share 19.3% identity. A prior study showed similar lattices in phage lysates from the *Salmonella* phage SPN3US, whose ChmA homolog shares only 10% sequence identity with 201Φ2-1 ChmA. Finally, a recent *bioRxiv* preprint (DOI: 10.1101/2022.04.06.487387) demonstrates the same finding with ΦPA3 ChmA. All three of these studies are now cited in this section of the revised manuscript (references 10, 11, and 28).

8. Line 193 onwards: The authors should explain why they are certain that the overexpression of wt or mutated variants of ChmB do not cause a growth defect that would affect subsequent observations with the pseudonucleus.

As is now shown in **Extended Data Fig. 7a-b**, expression of wild-type or mutant ChmB did not significantly impact the growth of uninfected cells.

Reviewer #3:

In this manuscript, the authors identify several proteins of jumbo phage PhiPA3 that interact with the ChmA protein, the major component of the pseudonucleus that compartmentalises phage DNA inside the infected cells and is the site of phage DNA replication and transcription. They show that at least one of the identified interactors of ChmA, a conserved jumbo [phage] protein they term ChmB, also binds to phage portal protein, suggesting that it may coordinate phage virion morphogenesis and DNA loading at the outer surface of the pseudonucleus. They present the high resolution structure of ChmB, a protein with a unique fold, and show that overproduction of ChmB with mutations in the conserved surface residues prevents pseudonucleus formation in phage-infected cells. Overproduction of wild-type ChmB leads to enlarged pseudonuclei.

These are interesting [observations] and [the] experiments are of high quality. Experiments of ChmB overproduction are potentially the most interesting for the understanding of pseudonucleus assembly and its role in phage progeny production. However, these experiments appear to be preliminary, since the functional consequences of overproduction of wild-type or mutant ChmB are not studied further. In particular, why are pseudonuclei increased in size upon wild-type protein overproduction? Why do nuclei fail to form upon mutant proteins overproduction? Are these direct or indirect effects? How is expression of phage early genes affected when cells overproducing ChmB are infected? How does the protein interaction network change upon ChmB overproduction? I believe these questions need to be addressed prior to publication in *NSMB*.

The reviewer correctly identifies a number of interesting questions arising from our identification of ChmB as a critical protein interaction hub of the jumbo phage nucleus. While we share the reviewer's curiosity about the answers to these questions, the experiments needed to answer them fall outside the scope of this manuscript. The current manuscript describes the protein interaction network of the jumbo phage nucleus, and the identification and characterization of ChmB's interactions with selected phage proteins (ChmA and the phage portal). As we note in the manuscript, we suspect that additional phage nucleus-associated factors we identified - gp61, gp63, and gp64 in particular - also associate with ChmB and likely play crucial roles in mRNA and protein translocation through the nuclear shell. Several of the reviewer's questions, including why the nucleus is larger when ChmB is overexpressed and why nucleus formation is inhibited when expressing ChmB mutants, hinge on a better understanding of these

proteins' interactions and roles. While we are actively pursuing these questions, they will require significant additional effort and are outside the scope of the current work.

Because of the imprecision of mass spectrometry as a quantitative analysis technique, we feel that any effects of ChmB mutations on the phage nuclear shell interaction network - using either Bio-ID or pulldowns - would be difficult to confidently interpret. Any changes we observe could easily be due to the "noise" in these analyses rather than true changes. Moreover, the presence of phage-encoded wild-type ChmB in cells expressing the mutant proteins would also complicate the analysis. We did show that mutant ChmB is partially compromised for ChmA binding (though not enough to strongly affect nuclear shell localization in infected cells), and that phage portal protein binding is unaffected (**Fig. 4f,g**).

Finally, while we recognize the reviewer's curiosity about how gene expression profiles change upon overexpression of ChmB, we feel that this analysis would also potentially result in ambiguous answers. For example, given the massive changes in phage nucleus morphology upon overexpression of wild-type or mutant ChmB, these changes will no doubt manifest as massive overall changes in phage gene expression. The most interesting question would be whether the timing of early, middle, and late gene expression is strongly affected, and if it is, what are the underlying mechanisms responsible for the changes. Understanding these questions will require an extensive analysis of how transcription by two different multisubunit virion RNA polymerases is regulated in vivo during the course of infection. We currently lack the synchrony and time resolution to perform these kinds of analyses with the required precision.

Decision Letter, first revision:**Message:** 9th Feb 2023

Dear Dr. Corbett,

Thank you again for submitting your manuscript "Identification of the bacteriophage nucleus protein interaction network". I sincerely apologize for the unusual delay in responding, which resulted from the difficulty in obtaining the referee reports. Nevertheless, we now have comments (below) from the 2 reviewers who originally evaluated your paper (reviewer #3 was no longer available for re-review). In light of those reports, we remain interested in your study and would like to see your response to the comments of the referees, in the form of a revised manuscript.

You will see that reviewer #2 emphasizes the need to validate that loss of direct interaction of gp2 with ChmA is what causes the observed effects on pseudo-nucleus formation through in vitro reconstitution of these proteins. Editorially, we agree with the reviewer that this would strengthen the conclusions of the manuscript. While we do not want to see manuscripts undergoing multiple rounds of revision, in this case we agree that this final point is important and want to see it addressed before we can make a final decision for publication at NSMB. Please be sure to address/respond to all concerns of the referees in full in a point-by-point response and highlight all changes in the revised manuscript text file. If you have comments that are intended for editors only, please include those in a separate cover letter.

We expect to see your revised manuscript within 6 weeks. If you cannot send it within this time, please contact us to discuss an extension; we would still consider your revision, provided that no similar work has been accepted for publication at NSMB or published elsewhere.

Reporting Summary:

When submitting the revised version of your manuscript, please pay close attention to our [href="https://www.nature.com/nature-portfolio/editorial-policies/image-integrity">Digital Image Integrity Guidelines. and to the following points below:](https://www.nature.com/nature-portfolio/editorial-policies/image-integrity)

Please note that all key data shown in the main figures as cropped gels or blots should be presented in uncropped form, with molecular weight markers. These data can be aggregated into a single supplementary figure item. While these data can be displayed in a relatively informal style, they must refer back to the relevant figures. These data should be submitted with the final revision, as source data, prior to acceptance, but you may want to start putting it together at this point.

Data availability: this journal strongly supports public availability of data. All data used in accepted papers should be available via a public data repository, or alternatively, as Supplementary Information. If data can only be shared on request, please explain why in your Data Availability Statement, and also in the correspondence with your editor. Please note that for some data types, deposition in a public repository is mandatory - more information on our data deposition policies and available repositories can be found below: <https://www.nature.com/nature-research/editorial-policies/reporting-standards#availability-of-data>

[REDACTED]

Sincerely,
Sara

Sara Osman, Ph.D.
Associate Editor
Nature Structural & Molecular Biology

Reviewers' Comments:

Reviewer #1:

Remarks to the Author:

I'm fully satisfied with authors' responses to my comments. I have no further comments.

Reviewer #2:

Remarks to the Author:

Enstun et al. provide a revised version of their manuscript which contains both text changes and new experiments. To recap, in this work the authors searched globally for new interaction partners of the pseudo-nucleus in phage-infected cells, a structure built primarily by the protein ChmA. The list of candidates was collapsed based on two main criteria: Co-localization with the pseudo-nucleus and pull-down of ChmA. Several candidates including the phage protein gp2 were selected for further study. The authors determined the structure of gp2 and predicted key residues for interaction with ChmA. In

order to validate the predicted importance of gp2 function in pseudo-nucleus formation and key role as a hub for other factors, the authors altered two of these proposed key residues of the gp2 protein to observe strong effects on pseudo-nucleus shape and formation (new Fig. 4b-c,e).

Major comment:

We have previously recommended the authors prove more directly the proposed loss of interaction between ChmA and the mutated gp2 proteins to support their claim that it really is the loss of interaction causes the observed effects on the pseudo-nucleus formation. The authors addressed this by a pull-down experiment (new Fig. 4f) where both interaction partners were co-expressed and ChmA was recovered after pull-down of gp2 and variants (similar to old Fig. 2e). While the authors interpret the results of this experiment to prove that the ChmA-gp2 is lost/diminished, the interpretation is not that straightforward at all. This is because ChmA protein levels were greatly reduced upon expression of the gp2 variants in the input. Thus, the conclusion that the gp2 variants are impaired for a direct molecular interaction with ChmA cannot be drawn from this experiment (new Fig. 4f). A validation of tagged ChmA in the inputs by western blot is missing. In addition, why do the gp2 variants drastically differ in molecular weight from the wild-type protein if only single amino acids were altered (new Fig. 4f,g)? This is particularly important because the only evidence for gp2 to influence pseudo-nucleus formation is based on these gp2 variants, so you want to be sure that these are the right proteins. Might they be post-translationally modified?

Ideally, the authors should validate that loss of direct interaction of gp2 with ChmA is what causes the observed effects on pseudo-nucleus formation. This could be done by performing *in vitro* reconstitution assays, checking whether ChmA oligomerization is different for the gp2 variants over the gp2 wild-type protein (could be done by size exclusion chromatography). The *in vitro* reconstitution experiment would be more informative than native pull-downs because the authors could better control for input levels of the ChmA and gp2 proteins. This is not an unreasonable request because the authors have already purified the gp2 variants; the ChmA protein is easy to purify as well.

If the authors cannot provide this type of experimental proof, they must refrain from overstating that the present results prove a direct interaction of ChmA and gp2 to serve in pseudo-nucleus formation.

Minor comments:

- Extended Data Fig. 4, problems with gene/protein identifiers: For example, gp63 in phiPA3 is not the correct identifier, YP_009217146, corresponds to gp64. And gp63 (YP_009217145.1) corresponds to PHIKZ068, which is the nvRNAP, whereas here gp62 was indicated as RNAP subunit but is a glycine-rich protein. In Table 3 gp62 is referred to as nvRNAP subunit but it should be gp63. The authors should carefully check their annotation again.
- The experiment shown in 4f is not an *in vitro* interaction, it represents *in vivo* expression and native pull-down.
- Fig. 4g: what is the band around 40 kDa that is only present with the wild-type protein?

Author Rebuttal, first revision:

Responses to Reviewers

We appreciate the insightful feedback from all reviewers during this process. We are gratified that Reviewer #1 is satisfied with the prior submitted version, and thank them for their assistance with strengthening the manuscript. Below, we address the remaining concerns of Reviewer #2 in detail. The Reviewer's points are in black text; our responses are in blue text. In the accompanying manuscript, all substantive changes from the prior submitted version are highlighted in red text.

Reviewer #2 Remarks to the Author:

Enustun et al. provide a revised version of their manuscript which contains both text changes and new experiments. To recap, in this work the authors searched globally for new interaction partners of the pseudo-nucleus in phage-infected cells, a structure built primarily by the protein ChmA. The list of candidates was collapsed based on two main criteria: Co-localization with the pseudo-nucleus and pull-down of ChmA. Several candidates including the phage protein gp2 were selected for further study. The authors determined the structure of gp2 and predicted key residues for interaction with ChmA. In order to validate the predicted importance of gp2 function in pseudo-nucleus formation and key role as a hub for other factors, the authors altered two of these proposed key residues of the gp2 protein to observe strong effects on pseudo-nucleus shape and formation (new Fig. 4b-c,e).

To clarify our logic, the gp2 (ChmB) point mutations we chose to mutate were never predicted to specifically disrupt interaction with ChmA or the phage portal protein (gp148). Rather, these mutations were chosen based on the residues' high conservation among ChmB orthologs in jumbo phages, and their location on the surface of the protein. As stated in the manuscript, "We generated point mutations of these residues (Q53A and A159D, respectively) designed to alter the ChmB surface and potentially disrupt specific protein-protein interactions". These mutations were also specifically designed to avoid disrupting the observed ChmB homodimer formation. While it was reasonable to expect that mutating conserved surface residues on ChmB might disrupt ChmA binding, our observations that the mutant ChmB proteins properly localize to the phage nucleus (as shown in the first submission) and maintain their direct interaction with ChmA in a pulldown assay (as shown in the current submission; see details below) clearly shows that a loss of ChmA binding is not the source of these mutants' phenotypic effects.

Major comment:

We have previously recommended the authors prove more directly the proposed loss of interaction between ChmA and the mutated gp2 proteins to support their claim that it really is the loss of interaction [that] causes the observed effects on the pseudo-nucleus formation.

Respectfully, this statement mischaracterizes both of our prior submissions' claims. As noted above, we never claimed that the phenotypic effects of expressing mutant ChmB proteins was caused by a loss of binding to ChmA. Indeed, our data showing that ChmB mutants successfully localize to the phage nuclear shell in infected cells (provided in the original submission) argues against this idea. Nonetheless, we did find the reviewer's suggestion that we directly test the mutants' effects on ChmA and portal protein binding

(from their first review: “The authors should perform pull down experiments with GFP-tagged mutated proteins and compare the results to those obtained for wt ChmB. Do these mutant proteins lose the ability to interact with ChmA or another interaction partner?”) compelling, prompting us to perform the pulldown assays described more fully below.

The authors addressed this by a pull-down experiment (new Fig. 4f) where both interaction partners were co-expressed and ChmA was recovered after pull-down of gp2 and variants (similar to old Fig. 2e). While the authors interpret the results of this experiment to prove that the ChmA-gp2 [interaction] is lost/diminished, the interpretation is not that straightforward at all. This is because ChmA protein levels were greatly reduced upon expression of the gp2 variants in the input. Thus, the conclusion that the gp2 variants are impaired for a direct molecular interaction with ChmA cannot be drawn from this experiment (new Fig. 4f). A validation of tagged ChmA in the inputs by western blot is missing. In addition, why do the gp2 variants drastically differ in molecular weight from the wild-type protein if only single amino acids were altered (new Fig. 4f,g)? This is particularly important because the only evidence for gp2 to influence pseudo-nucleus formation is based on these gp2 variants, so you want to be sure that these are the right proteins. Might they be post-translationally modified?

We thank the reviewer for their keen eye with regard to the coexpression pulldown assays presented in the former **Figure 4f-g** (now **Extended Data Figure 7d-e**). First, the differing apparent molecular weights of gp2 point mutants compared to wild type is very common for point mutants that alter a protein’s overall charge/hydrophobicity characteristics, especially with small proteins such as gp2. Specifically, a protein’s migration on SDS-PAGE depends both on its inherent charge and on its ability to bind the SDS detergent in the denatured state. We nonetheless verified that all clones are correct by sequencing. The proteins are unlikely to be post-translationally modified, since these experiments are done by coexpression in *E. coli*.

The reviewer makes a good point about the differing input levels of ChmA in our pulldown assay presented in the former **Figure 4f** (now **Extended Data Figure 7d**). Spurred by this comment, we performed the same pulldown assay again in triplicate, and now observe no significant difference in ChmA binding between wild-type gp2 (ChmB) and the two point mutants. This observation agrees with our finding that the ChmB point mutants also localize to the phage nuclear shell in infected cells (**Figure 4a**, **Figure 4e**, and **Extended Data Figure 8**). We thank the reviewer for catching this problem with our prior submission.

Ideally, the authors should validate that loss of direct interaction of gp2 with ChmA is what causes the observed effects on pseudo-nucleus formation. This could be done by performing in vitro reconstitution assays, checking whether ChmA oligomerization is different for the gp2 variants over the gp2 wild-type protein (could be done by size exclusion chromatography). The in vitro reconstitution experiment would be more informative than native pull-downs because the authors could better control for input levels of the ChmA and gp2 proteins. This is not an unreasonable request because the authors have already purified the gp2 variants; the ChmA protein is easy to purify as well.

The reviewer's thinking strongly parallels our own. We did seek to reconstitute a stable ChmA-gp2 complex *in vitro* to examine the effects of gp2 on ChmA oligomerization, and by extension how gp2 mutants might affect this behavior. After repeated trials, we observe that *in vitro*, the ChmA-gp2 interaction is not stable. The two proteins do not interact with one another when separately purified and then mixed. Even when coexpressed in *E. coli* (as in **Extended Data Figure 7d**), the complex is not stable and the proteins dissociate from one another through the course of a multi-step purification. We believe that this behavior is due to a combination of ChmA's distinctive self-oligomerization properties and the requirements for gp2-ChmA association: When expressed in *E. coli*, ChmA adopts three oligomeric states (described in PMID 35922510): monomers, stable oligomers comprising six ChmA tetramers (24 copies total), and larger oligomers that mimic the native structure of the ChmA shell (albeit forming much smaller assemblies than that formed in native infections). We believe that gp2 interacts solely with oligomeric ChmA in the "shell" form (based on the results in **Figure 2e**), integrating into the lattice and displacing one or more ChmA protomers (possibly a full tetramer as outlined in **Figure 5b**). We believe that *in vitro*, the energetics favor the assembly of highly stable ChmA lattices and that this tends to eject gp2 from the ChmA lattice over time. For similar energetic reasons, gp2 cannot enter an already-assembled stable ChmA lattice (as in an *in vitro* mixing experiment). Therefore, the pulldown assays now presented in **Extended Data Figure 7d-e** are the most we can do to test the effects of gp2 mutations on ChmA binding.

If the authors cannot provide this type of experimental proof, they must refrain from overstating that the present results prove a direct interaction of ChmA and gp2 to serve in pseudo-nucleus formation.

We respectfully disagree that the current set of data does not support a direct interaction between gp2 (ChmB) and ChmA. ChmB was identified in our Bio-ID screens with ChmA (**Table 1**); ChmB shows strong nuclear shell localization in infected cells (**Figure 1d**); ChmA was a strong hit in our pulldowns of GFP-tagged ChmB from native infections (**Figure 2c-d**); and the two proteins interact with one another when expressed in *E. coli* without any other phage or host proteins present (**Figure 2e**). Our pulldown assays are only one of several lines of evidence, all of which support the conclusion that these proteins directly interact with one another.

Moreover, as noted above we never claimed that the observed effects of ChmB mutations on pseudo-nucleus formation and the progression of infections is specifically due to their effects on ChmA binding. We were careful, in fact, to not make this strong claim in either of our prior submissions. We explicitly noted, for example, that the ChmB mutations do not compromise the protein's localization to the nuclear shell in infected cells (**Figure 4e, Extended Data Fig. 8**), meaning that the protein still interacts with ChmA during infections. Our new pulldowns showing that ChmB point mutations do not disrupt ChmA binding in coexpression assays further underscore this point, but do not change the significance of our observations with respect to these mutations' effects on jumbo phage infections. ChmB is a multifunctional protein whose various roles are just beginning to be explored. In this light, further experimentation will be required to fully understand ChmB's roles in infections, and by extension how our mutants affect these different roles. These points are made clearly in the **Discussion**.

Minor comments:

Extended Data Fig. 4, problems with gene/protein identifiers: For example, gp63 in phiPA3 is not the correct identifier, YP_009217146, corresponds to gp64. And gp63 (YP_009217145.1) corresponds to PHIKZ068, which is the nvRNAP, whereas here gp62 was indicated as RNAP subunit but is a glycine-rich protein. In Table 3 gp62 is referred to as nvRNAP subunit but it should be gp63. The authors should carefully check their annotation again.

This confusion arises from the fact that the genome record for PhiPA3 (NCBI RefSeq NC_028999.1) has two sets of numbering for most gene products. For example, NCBI # YP_009217145.1 is annotated as gene "062" and locus tag "AVT69_gp063" (see attached screenshot from the NCBI page describing YP_009217145.1). We are using the "gene" identifier for all gene products rather than the locus tag: for this protein, we therefore term it "gp62". We recognize that this is somewhat confusing, but this is due to inconsistency within NCBI's records, not our own error. Screen shot from NCBI's record for YP_009217145.1 (note the section outlined in red):

```
FEATURES             Location/Qualifiers
    source             1..525
                     /organism="Pseudomonas phage PhiPA3"
                     /host="Pseudomonas aeruginosa strain PA01"
                     /db_xref="taxon:998086"
    Protein            1..525
                     /product="hypothetical protein"
                     /calculated_mol_wt=59572
    CDS                1..525
                     /gene="062"
                     /locus_tag="AVT69_gp063"
                     /coded_by="NC_028999.1:48722..50299"
                     /note="similar to gp124 in Pseudomonas phage 201phi2-1,
                     gp068 Pseudomonas phage phiKZ"
                     /transl_table=11
                     /db_xref="GeneID:26643593"
ORIGIN
   1 mqitvtgvqg  sgfvectish  kgqsitwstr  ayskvklqdp  trvfkeindy  leyagedvqd
  61 kiwneykrir  dlnmfdpds  hitatlihyi  qqiysvlpmn  nmrrwlltvg  nlhipadiqk
 121 titedsrynk  reqtlqhdy  inlatvalal  rvmipiwgey  mdqgtdqdly  rendvvglis
 181 rcevanwpm  etdtvgepvd  tafdklegyv  rfcvedeptt  lgrlwsqmss  veipvhlqsk
 241 vlvrlltipv  lndpsshsiv  anvfryvrsn  lnpterstad  rvnekrpegg  sgdeddktfs
 301 ieahkktgrv  spgdieafni  damdyellae  tvdptinkam  lkacincidy  iankeirphq
 361 vllaqwvmak  afparafyhi  dkmpvnhlla  ttqallwhwg  fldvaifmqv  eplyghdhs
 421 snqlsqrsg  arianrykpm  ldelyphmkl  akvpqsgeip  kpdnmagiai  nsanasirss
 481 nwfvrppel  fnaanqvaqn  dvlmvpnik  haltemvmhl  akinq
//
```

To ensure that this confusion does not result in any misinterpretation of our data, we have added NCBI identifiers in a new column in **Tables 1-3** and **Extended Data Tables 1-4** (these identifiers were already present in **Extended Data Table 6**).

The experiment shown in 4f is not an in vitro interaction, it represents in vivo expression and native pull-down.

We previously used the term "in vitro" for this assay since the proteins were being studied outside their native environment. We recognize that since the proteins were coexpressed in an *E. coli* cell, the term "in vitro" could be confusing. Therefore, we have

replaced the term “in vitro” with “pulldown assay” when referring to the experiment described in the former **Figure 4f** (now **Extended Data Figure 7d**).

Fig. 4g: what is the band around 40 kDa that is only present with the wild-type protein?

This is an unknown contaminant. We now mention this band in the figure legend.

Decision Letter, second revision:

Message: Our ref: NSMB-A46448B

22nd Mar 2023

Dear Dr. Corbett,

Thank you for submitting your revised manuscript "Identification of the bacteriophage nucleus protein interaction network" (NSMB-A46448B). The revised manuscript and your detailed responses to the outstanding referee concerns have now been considered by the editorial team, and we have decided that we'll be happy in principle to publish your manuscript in Nature Structural & Molecular Biology, pending minor revisions to comply with our editorial and formatting guidelines.

We are now performing detailed checks on your paper and will send you a checklist detailing our editorial and formatting requirements in a couple of weeks. Please do not upload the final materials and make any revisions until you receive this additional information from us.

To facilitate our work at this stage, it is important that we have a copy of the main text as a word file. If you could please send along a word version of this file as soon as possible, we would greatly appreciate it; please make sure to copy the NSMB account (cc'ed above).

Sincerely,
Sara

Sara Osman, Ph.D.
Associate Editor
Nature Structural & Molecular Biology

Final Decision Letter:

Message 11th Aug 2023

:

Dear Dr. Corbett,

We are now happy to accept your revised paper "Identification of the bacteriophage nucleus protein interaction network" for publication as a Article in Nature Structural & Molecular Biology.

Your paper will be published online soon after we receive proof corrections and will appear in print in the next available issue. You can find out your date of online publication by contacting the production team shortly after sending your proof corrections. Content is published online weekly on Mondays and Thursdays, and the embargo is set at 16:00 London time (GMT)/11:00 am US Eastern time (EST) on the day of publication. Now is the time to inform your Public Relations or Press Office about your paper, as they might be interested in promoting its publication. This will allow them time to prepare an accurate and satisfactory press release. Include your manuscript tracking number (NSMB-A46448C) and our journal name, which they will need when they contact our press office.

About one week before your paper is published online, we shall be distributing a press release to news organizations worldwide, which may very well include details of your work. We are happy for your institution or funding agency to prepare its own press release, but it must mention the embargo date and Nature Structural & Molecular Biology. If you or your

Press Office have any enquiries in the meantime, please contact press@nature.com.

Please note that *Nature Structural & Molecular Biology* is a Transformative Journal (TJ). Authors may publish their research with us through the traditional subscription access route or make their paper immediately open access through payment of an article-processing charge (APC). Authors will not be required to make a final decision about access to their article until it has been accepted. [Find out more about Transformative Journals](https://www.springernature.com/gp/open-research/transformative-journals)

Sincerely,

Carolina Perdigoto, PhD
Chief Editor
Nature Structural & Molecular Biology
orcid.org/0000-0002-5783-7106

Click here if you would like to recommend Nature Structural & Molecular Biology to your librarian:

<http://www.nature.com/subscriptions/recommend.html#forms>